# Loquetier: A Virtualized Multi-LoRA Framework for Unified LLM Fine-tuning and Serving

Yuchen Zhang[1]    Hanyue Du[1]    Chun Cao[1]    Jingwei Xu[1][*]

[1]State Key Laboratory for Novel Software Technology, Nanjing University, China
{zfirozen,dhy}@smail.nju.edu.cn, {caochun,jingweix}@nju.edu.cn

## Abstract

Low-Rank Adaptation (LoRA) has become a widely adopted parameter-efficient fine-tuning (PEFT) technique for adapting large language models (LLMs) to downstream tasks. While prior work has explored strategies for integrating LLM training and serving, there still remains a gap in unifying fine-tuning and inference for LoRA-based models. We present **Loquetier**, a virtualized multi-LoRA framework that seamlessly integrates LoRA fine-tuning and serving within a single runtime. Loquetier introduces two key components: (1) a *Virtualized Module* that isolates PEFT-based modifications and supports multiple adapters on a shared base model, and (2) an optimized computation flow with a kernel design that merges fine-tuning and inference paths in forward propagation, enabling efficient batching and minimizing kernel invocation overhead. Extensive experiments across three task settings show that Loquetier consistently outperforms existing baselines in both performance and flexibility, achieving up to $3.0\times$ the throughput of the state-of-the-art co-serving system on inference-only tasks and $46.4\times$ higher SLO attainment than PEFT on unified fine-tuning and inference tasks. The implementation of Loquetier is publicly available at `https://github.com/NJUDeepEngine/Loquetier`.

## 1   Introduction

Large Language Models (LLMs) built on stacked transformer blocks [Vaswani et al., 2017] have achieved remarkable success across a wide range of text generation tasks. This success has driven the development of ever-larger models, such as the LlaMA series [Touvron et al., 2023, Grattafiori et al., 2024] and the Qwen family [Bai et al., 2023, Yang et al., 2024]. However, the rapid growth in model size has introduced prohibitive costs. For example, LlaMA 3 contains 405B parameters, and DeepSeek-V3 [Bi et al., 2024] scales to 671B. Their computational and memory requirements of full-parameter training now represent a major bottleneck, restricting both the scalability and accessibility of LLM development.

Parameter-efficient fine-tuning (PEFT) has emerged as a practical solution to these challenges. By reducing the number of trainable parameters while retaining the effectiveness of full-model fine-tuning, PEFT offers a balance between efficiency and adaptability [Ding et al., 2023a]. Recent studies have evaluated PEFT across diverse applications and theoretical dimensions [Balne et al., 2024, Xu et al., 2023a], and have surveyed its underlying mechanisms and practical benefits [Han et al., 2024, Fu et al., 2023]. Notice that PEFT approaches such as Prefix Tuning [Li and Liang, 2021] and Prompt Tuning [Lester et al., 2021] have demonstrated strong adaptability by optimizing small task-specific vectors. Moreover, PEFT has shown competitive or superior performance in low-resource settings, including zero- and few-shot learning [Liu et al., 2022, Hu et al., 2023].

---

[*]Corresponding author

39th Conference on Neural Information Processing Systems (NeurIPS 2025).

Table 1: Comparison on different LoRA tasks between Loquetier, PEFT and FlexLLM

| Framework or System | Inference | | Finetune | | Finetune & Inference | |
|---|---|---|---|---|---|---|
| | Single | Multi | Single | Multi | Single | Multi |
| Loquetier | ✓ | ✓ | ✓ | ✓ | ✓ | ✓ |
| PEFT | ✓ | ✓ | ✓ | × | ✓ | × |
| S-LoRA+PEFT | ✓ | ✓ | ✓ | × | ✓ | × |
| FlexLLM | ✓ | △[3] | ✓[4] | ×[4] | × | × |

Among PEFT approaches, *Low-Rank Adaptation* (LoRA) [Hu et al., 2022] has become particularly prominent, offering scalable and effective fine-tuning across diverse LLM tasks. Numerous extensions have been proposed to enhance its flexibility, including LoHa [Hyeon-Woo et al., 2021], VeRA [Kopiczko et al., 2023], and LoKr [Yeh et al., 2023]. LoRA has also shown strong potential for personalization, powering systems in recommendation [Kong et al., 2024, Zhu et al., 2024], and user-centered content generation [Zhang et al., 2024, Wu et al., 2024a]. Its modularity makes it well-suited for large-scale serving systems where real-time customization is critical.

In practice, systems that serve LoRA often need to support fine-tuning adapters while simultaneously deploying them for inference across diverse tasks. However, no existing framework can seamlessly unify these two capabilities, resulting major obstacles for scaling LoRA into production. Jointly fine-tuning and serving multiple adapters requires minimizing memory and computation overhead while efficiently handling heterogeneous workloads. Prior efforts have mainly optimized the base LLM, such as through KVCache improvements or inference pipeline parallelization [Li et al., 2024a], with only limited advances in multi-LoRA inference [Chen et al., 2024, Sheng et al., 2023] or integrated fine-tuning and inference [Miao et al., 2024]. However, these approaches still face critical challenges: adapters are often fused into monolithic instances for efficiency, thus they cannot be dynamically loaded or unloaded; decoding efficiency degrades significantly when fine-tuning and inference run concurrently; and task switching typically requires halting the current job before starting another, causing downtime, bandwidth overhead, and wasted resources. As a result, existing frameworks remain inadequate for large-scale, production-ready LoRA applications.

In this paper, we introduce **Loquetier**[2], a unified virtualization framework that integrates fine-tuning and serving of LLMs with LoRA-based PEFT. Loquetier provides a streamlined computation flow that handles both fine-tuning and inference requests within a shared runtime, using kernel-level optimizations to reduce memory overhead and execution latency. Specifically, Loquetier introduces the *Segmented Multi-LoRA Multiplication (SMLM)* kernel, which enables mixed-task execution by distinguishing forward-pass behaviors for fine-tuning, evaluation, prefilling, and decoding. To support multiple concurrent LoRA adapters without modifying the base model, Loquetier further incorporates a *Virtualized Module* abstraction that dynamically injects adapter logic while preserving compatibility and isolation across tasks and devices. This design enables seamless co-serving of heterogeneous LoRA configurations and supports instance-to-instance migration of fine-tuning jobs without kernel restarts or memory duplication. The main contributions are as follows:

- We design an SMLM kernel and an unified computation flow that efficiently supports fine-tuning and inference with multiple LoRA adapters on a shared base model.

- We propose a modular virtualization mechanism that isolates PEFT-based modifications from the base model, enabling flexible instance-level migration and seamless adapter management.

- We develop Loquetier to unify LoRA fine-tuning and serving. Extensive experiments show that Loquetier outperforms existing systems across diverse scenarios and enables unified practical fine-tuning and serving configurations previously unsupported.

---

[2]The name *Loquetier* is a synthesis of "LoRA" and "coquetier", reflecting our design philosophy: the base model serves as the foundational spirit, while LoRA modules act as adjunct ingredients-spirits, juices, and syrups-mixed in for task-specific customization.

[3]FlexLLM cycles through loading LoRA models during multi-LoRA inference, disregarding the maximum number of resident LoRAs set, which makes its multi-LoRA inference efficiency practically unusable.

[4]The backward procedure of FlexLLM triggered an error originating from an unsupported operation in its gradient computation logic. We describe our solution in the Appendix B.

## 2 Related Work

**LLM inference optimization via KV cache.** Key-value (KV) caching is a core technique for accelerating LLM inference by avoiding redundant computation and memory transfers [Pope et al., 2023]. Recent work improves cache efficiency and reduces GPU memory overhead through tailored management strategies. Prompt Cache [Gim et al., 2024] reuses prompt embeddings to reduce duplication. vLLM [Kwon et al., 2023] and vAttention [Prabhu et al., 2025] address fragmentation using paging and virtual memory, respectively. Infinite-LLM [Lin et al., 2024] enables dynamic sharing of cache segments between host and GPU. LoongServe [Wu et al., 2024b] enhances long-context serving by balancing prefilling and decoding workloads.

**LLM inference parallelism.** A large amount of work improves LLM inference throughput by optimizing batch scheduling and pipeline execution. Response Length Perception [Zheng et al., 2023] and $S^3$ [Jin et al., 2023] predict output lengths to batch similar requests, with $S^3$ further refining its predictions over time. Orca [Yu et al., 2022] uses token-level continuous batching, while DeepSpeed-Fastgen [Holmes et al., 2024] adjusts request lengths for better GPU utilization. Sarathi-Serve [Agrawal et al., 2024] co-schedules prefilling and decoding tasks, whereas TetriInfer [Hu et al., 2024], Splitwise [Patel et al., 2024], and DistServe [Zhong et al., 2024] decouple these phases across threads, machines, or clusters to improve parallel efficiency. In kernel, FlashDecoding++ [Hong et al., 2024] overlaps computation with data transfer to hide latency. FlashAttention [Dao, 2023, Shah et al., 2024] uses warp specialization and asynchronous execution to maximize attention-layer throughput.

**Efficient multi-LoRA inference and unified fine-tuning-inference systems.** FlashInfer [Ye et al., 2025] reduces redundant storage and optimizes GPU memory access for efficient multi-LoRA inference. Cutlass [Thakkar et al., 2023], a high-performance CUDA library for GEMM operations, serves as the kernel foundation for Punica [Chen et al., 2024], which combines with FlashInfer to support scalable LoRA serving. Built on this stack, S-LoRA [Sheng et al., 2023] further improves GPU memory utilization through dynamic host-device memory transfers. In parallel, FlexLLM [Miao et al., 2024] explores unified token-level computation for co-serving inference and PEFT fine-tuning, though scalability remains limited.

**LoRA optimization and variants.** There are lots of extensions that have been proposed to enhance the flexibility, efficiency, and adaptability of LoRA. AdaLoRA [Zhang et al., 2023a] introduces singular value decomposition for dynamic pruning, while IncreLoRA [Zhang et al., 2023b] allocates parameters based on module importance. SoRA [Ding et al., 2023b] adaptively adjusts rank via gated weight control during training. DiffoRA [Jiang et al., 2025] selects modules to finetune using a Differential Adaptation Matrix (DAM). To improve expressiveness and stability, DoRA [Liu et al., 2024] decomposes weights into magnitude and direction. VB-LoRA [Li et al., 2024b] reduces redundancy by sharing global vector banks across modules. LoRA-XS [Bałazy et al., 2024] compresses storage with minimal $r \times r$ matrices, and QA-LoRA [Xu et al., 2023b] integrates quantization and grouping to increase representational flexibility.

## 3 Loquetier Framework

In this section, we describe the overall architecture of Loquetier, followed by details of its unified computation flow and the proposed LoRA model virtualization.

### 3.1 Framework

As illustrated in Figure 1, Loquetier framework consists of two core components: (1) a model library centered around the *Virtualized Module* for isolating PEFT modifications, and (2) a redesigned kernel and computation flow that jointly support fine-tuning and inference workloads. To enable concurrent execution, Loquetier integrates a context management system that coordinates the runtime scheduling of heterogeneous tasks.

When loading a base model into CPU or GPU memory, Loquetier instantiates multiple *virtual models*, each acting as an isolated container for a specific PEFT configuration. These virtual models are bound to distinct adapters, enabling independent and concurrent execution. For LoRA-based adapters, we introduce the `MixedLoraModel` class to support fine-tuning. Based on our computation flow (see Section 3.3), each `MixedLoraModel` efficiently fine-tunes its associated LoRA adapter within

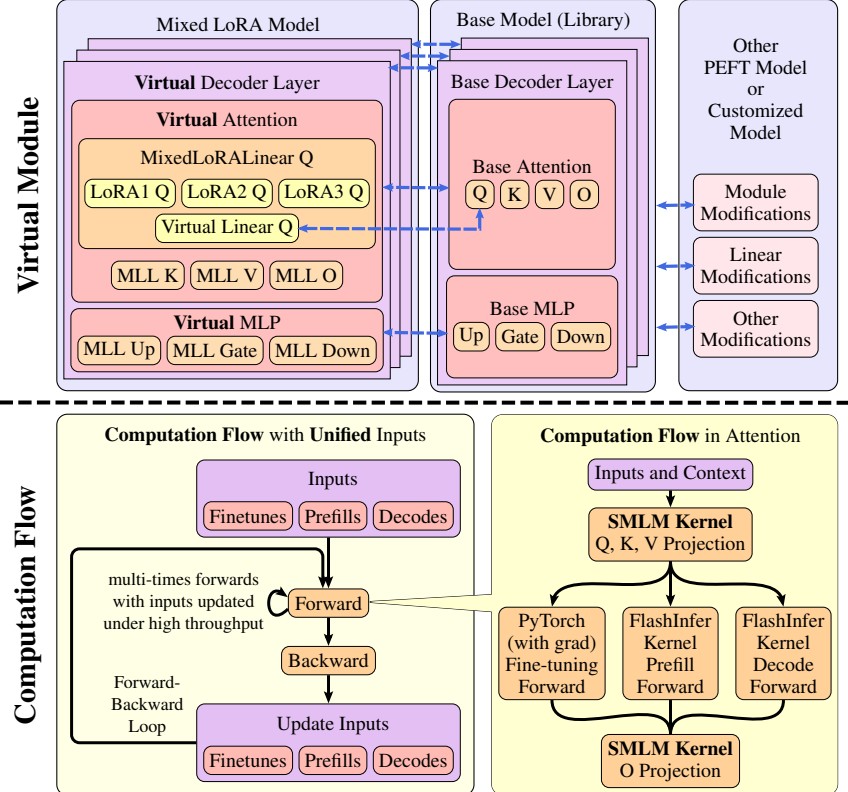

Figure 1: The framework diagram of Loquetier.

its own container from dedicated trainers, while other virtual models can simultaneously remain in inference mode.

Loquetier's modular design is compatible with most existing LLM inference optimizations and training strategies. It fully supports architectures that leverage FlashInfer [Ye et al., 2025] kernel and remains extensible to those that do not, requiring only localized modification to the inference logic within the computation flow. Section 3.2 further details how the Virtualized Module enables base model sharing across diverse PEFT methods beyond LoRA.

## 3.2 Virtualized module

The mixing of different LoRA or other PEFT methods in the same model object makes model configurations chaotic and difficult to handle. Moreover, dynamic model loading and unloading should be supported in order to apply the fine-tuned and up-to-date LoRA models quickly.

We propose the Virtualized Module that provides methods and data proxies to foundation modules to solve the above problems. Applying the Virtualized Module to the base model is extremely low-cost, with no additional GPU memory overhead and provides independent model instances in an intuitive way. For each module type, the Virtualized Module creates the corresponding virtual module at runtime to provide the correct proxy methods. The Virtualized Module prepares additional properties and methods for Linear and Model to accommodate the architecture of Transformers and PEFT.

Since each virtual module class is created at runtime, which means none of these virtual modules can be shared, nor can a process that owns any virtual module create child processes from fork or spawn method. Furthermore, any deep copying behavior that includes a virtual module will cause the base module linked to it to be copied, thus invalidating base module sharing. For this reason, we provide a non-local class definition for deep copying, serialization, and deserialization. By voiding the containing Virtualized Module, LoRA models and other PEFT models loaded onto the Virtualized

**Algorithm 1** Computation flow control in attention layer of Loquetier

---

**Input:** hidden states matrix $\mathbf{X}$ with shape $[S, H]$, list of fine-tuning inputs batch-sequence information tuples $\mathbf{F}$, list of prefilling and evaluation inputs sequence lengths $\mathbf{P}$, decoding inputs count $\mathbf{D}$.

**Output:** attention outputs matrix $\mathbf{O}$.
    $\mathbf{Q} = Q_{proj}(\mathbf{X}); \mathbf{K} = K_{proj}(\mathbf{X}); \mathbf{V} = V_{proj}(\mathbf{X});$
    $\mathbf{Os} = [];$
    **if** $len(\mathbf{F}) > 0$ **then**
        Extract $\mathbf{Q_f}, \mathbf{K_f}, \mathbf{V_f}$ from $\mathbf{Q}, \mathbf{K}, \mathbf{V}$ based on $\mathbf{F}$;
        Compute $\mathbf{O_f}$ through the standard forward implementation;
        $\mathbf{O_f}$ is appended to $\mathbf{Os}$;
    **end if**
    **if** $len(\mathbf{P}) > 0$ **then**
        Compute offset of prefills $\mathbf{Offset_p}$ based on $\mathbf{F}$;
        Extract $\mathbf{Q_p}, \mathbf{K_p}, \mathbf{V_p}$ from $\mathbf{Q}, \mathbf{K}, \mathbf{V}$ based on $\mathbf{P}$ and $\mathbf{Offset_p}$;
        Initialize KVCache for prefills;
        Compute $\mathbf{O_p}$ through the FlashInfer forward implementation;
        $\mathbf{O_p}$ is appended to $\mathbf{Os}$;
    **end if**
    **if** $\mathbf{D} > 0$ **then**
        Compute offset of decodes $\mathbf{Offset_d}$ based on $\mathbf{F}$ and $\mathbf{P}$;
        Extract $\mathbf{Q_d}, \mathbf{K_d}, \mathbf{V_d}$ from $\mathbf{Q}, \mathbf{K}, \mathbf{V}$ based on $\mathbf{Offset_d}$;
        Append KVCache for decodes;
        Compute $\mathbf{O_d}$ through the standard forward implementation;
        $\mathbf{O_d}$ is appended to $\mathbf{Os}$;
    **end if**
    Concatenate $\mathbf{Os}$ into one tensor as $\mathbf{O}$;
    $\mathbf{O} = O_{proj}(\mathbf{O});$
    return $\mathbf{O}$;

---

Module can be migrated to other GPU devices after deep-copying and used after unvoiding based on instances of the new Virtual Model.

Based on the above design, the Virtualized Module is compatible with all PEFT methods, as well as any custom model modification methods that do not modify the underlying model's own data, such as weights and module configurations. For scenarios where the target module overwrites the base module with new data, the module design needs to be normalized so that the target module runs the new forward method based on its own data and the base module's data, rather than running the base module's forward method after using destructive modification methods.

### 3.3 Unified computation flow management and SMLM kernel

For multiple LoRA adapters within the same linear layer, traditional methods typically process the computation sequentially, computing the output for one LoRA at a time and iterating through all adapters. This approach significantly slows computation. Since different LoRAs within the same layer usually share identical or similar shapes, it is feasible to compute all outputs in a single kernel call. Punica [Chen et al., 2024] leverages this property by implementing a foundational algorithm on top of the Cutlass GEMM (General Matrix Multiplication) library [Thakkar et al., 2023], enabling simultaneous processing of multiple input-LoRA pairs for efficient multi-adapter computation.

However, the original Punica kernel design is incompatible with fine-tuning, as it statically concatenates LoRA weights from the same module across different layers. This rigid coupling limits architectural flexibility and prevents selective application of LoRA to specific layers, which is particularly problematic in training scenarios where layerwise heterogeneity is common. This requires every layer in fine-tuning tasks to adopt the exact same configuration, even when certain linear modules do not actually require LoRA. Meanwhile, this design incurs substantial overhead in tracking and memory management, especially when operating on very large tensors during training. For inference tasks,

**Algorithm 2** Computation flow control in causal LM of Loquetier

---

**Input:** LM inputs $\mathbf{X}$, list of fine-tuning inputs batch-sequence information tuples $\mathbf{F}$, list of evaluation inputs sequence lengths $\mathbf{E}$, Labels of fine-tuning and evaluation inputs $\mathbf{Labels}$, Accumulation steps of fine-tuning and evaluation inputs $\mathbf{A}$.

**Output:** list of losses $\mathbf{Loss}$, logits $\mathbf{Logits}$.

    Compute $\mathbf{Logits}$ by forward propagation;

    $\mathbf{Loss} = []$

    **for** (batch-sequence $(\mathbf{B}, \mathbf{S})$, accumulation step $\mathbf{A_{FE}}$ in $(\mathbf{F}, \mathbf{E})$, $\mathbf{A}$) **do**

        Extract $\mathbf{Logits_{FE}}$ and $\mathbf{Labels_{FE}}$ from $\mathbf{Logits}$ and $\mathbf{Labels}$;

        Shift $\mathbf{Logits_{FE}}$ and $\mathbf{Labels_{FE}}$;

        Compute $\mathbf{Loss_{FE}}$ of $\mathbf{Logits_{FE}}$ and $\mathbf{Lables_{FE}}$ from the given loss function;

        $\mathbf{Loss_A} = \mathbf{Loss_{FE}}/\mathbf{A_{FE}}$;

        $\mathbf{Loss_A}$ is appended to $\mathbf{Loss}$;

    **end for**

    return $\mathbf{Loss}, \mathbf{Logits}$;

---

it also eliminates the possibility of rapidly swapping LoRAs during runtime. Thus, the computation process must be halted before replacement, and the required LoRA must be re-spliced together.

To overcome these limitations, we adapt the Punica kernel to process LoRA weights one linear layer at a time. This decoupling removes the need to regenerate model files via weight transformation before execution. For static scaling factors in LoRA, we apply the scale directly to the weight tensor at `MixedLoraModel` instantiation to reduce runtime overhead. When dynamic scaling is required, it is applied on a per-request basis during the forward pass. Loquetier then executes forward computation across all active requests in a unified manner, supporting four types of requests: fine-tuning (training), evaluation, prefilling, and decoding. Evaluation requests are structurally similar to prefilling but compute a loss over labels and execute only a single generation pass, while prefilling requests omit the loss computation and transition into decoding after the initial pass.

For the backward pass, since FlashInfer does not support gradient computation, Loquetier falls back to the standard forward implementation backed by PyTorch's Autograd when handling fine-tuning requests. This enables full gradient computation through efficient C++-based differentiation. For inference-only requests, Loquetier instead leverages FlashInfer's batch-prefill and batch-decode kernels to maximize throughput and memory efficiency.

The forward computation flow for the attention layer is summarized in Algorithm 1. First, the $\mathbf{Q}$, $\mathbf{K}$, and $\mathbf{V}$ projections are computed jointly for all incoming requests. Attention outputs are then computed independently for each request type, concatenated, and passed through a shared output projection $\mathbf{O}$. Algorithm 2 outlines the forward method for the causal language model. For each incoming request, Loquetier computes output logits and, when applicable, the loss with respect to provided labels. Fine-tuning and evaluation requests include ground truth labels and thus return both logits and loss values, while prefilling and decoding requests return only logits. Loquetier enables joint forward pass of fine-tuning and evaluation requests within the same batch. Because the losses are tracked separately in Loquetier, this separation allows distinct gradient accumulation strategies for different fine-tuning tasks in parallel, without cross-interference. By summing losses across all fine-tuning requests, Loquetier produces a shared backward pass, enabling gradients from multiple fine-tuning jobs to be computed efficiently in a single backpropagation step.

We further extend the Transformers' Trainer [Wolf et al., 2020] to support an interruptible fine-tuning process. Multiple trainers can now share the same computation flow in Loquetier, performing unified forward and backward propagation for fine-tuning different LoRA adapters concurrently. To ensure that each trainer only updates its corresponding parameters, we introduce `MixedLoRAModelForTrainer`, which applies parameter masking on top of a shared `MixedLoraModel` instance to achieve isolation.

## 4 Experiments

The Loquetier framework expects to enable fine-tuning and reasoning across multiple LoRA models. Based on the objective, we design three experiments to evaluate the performance of Loquetier in

Table 2: Comparison on model loading between Loquetier, PEFT, S-LoRA and FlexLLM. Metrics include time to load (Time) and additional storage footprint (Storage).

| Framework or System | Base Model | | LoRA Model | | Total | |
|---|---|---|---|---|---|---|
| | Time | Storage | Time | Storage | Time | Storage |
| Loquetier | 2.927 s | 0 B | 2.409 s | 0 B | 5.336 s | 0 B |
| PEFT | 2.877 s | 0 B | 1.914 s | 0 B | 4.791 s | 0 B |
| S-LoRA | 33.037 s | 0 B | 0.948 s | 0 B | 33.985 s | 0 B |
| FlexLLM | 37.933 s | 14.96 GB | 0.924 s | 40.04 MB | 38.857 s | 15.00 GB |

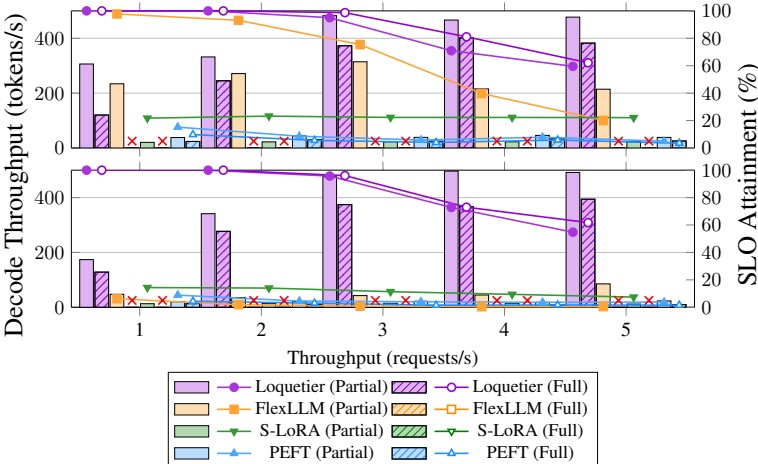

Figure 2: Comparison of the performance of Loquetier, FlexLLM, S-LoRA and PEFT in inference tasks. The upper is single LoRA model inference and the lower part is multiple LoRA model inference. Partial means that only 3 modules are enabled for FlexLLM including up, gate, and down. For detailed information on S-LoRA, please refer to the Appendix E. Full means that all 7 modules are enabled, including q, k, v, o, up, gate, and down. × indicates that the results were not obtained: FlexLLM does not support enabling LoRA modules for linear layers other than up, gate, and down; FlexLLM cycles through loading LoRA models during multi-LoRA inference.

these scenarios: inference-only, fine-tuning-only, and unified fine-tuning and inference. In addition, we perform two experiments to evaluate the performance of Loquetier in simulated real-world environments: a shorter, approximate simulation for rapidly testing different load conditions across different times of the day, and a 120-minute precise simulation using data extracted from real-world scenarios for a more demanding and realistic stress test. To further validate the effectiveness of our design, we conduct a micro-experiment to evaluate the efficiency of the model loading process.

## 4.1 Evaluation settings

**Baselines**. FlexLLM is an advanced co-serving system for LLM serving and parameter-efficient fine-tuning. We deployed docker images as its runtime environment following the guidelines. S-LoRA is designed specifically for large-scale LoRA inference. Therefore, we combine S-LoRA with PEFT as another baseline, in which PEFT handles the fine-tuning task. We use HuggingFace Transformers with PEFT as the most basic baseline.

**Models**. We use the Llama3-8B model as the base model. The LoRA adapter is obtained by training on the Alpaca dataset. For the fine-tuning task, we use the same LoRA configuration as the inference LoRA adapter and initialize the weights from the Gaussian distribution.

**Datasets**. We use the ShareGPT dataset as input for the inference task, and the Alpaca and GSM8K datasets as input for the fine-tuning task. BurstGPT [Wang et al., 2025] is an LLM service workload dataset comprising over ten million traces collected from Azure OpenAI GPT services. Data spanning more than 60 days is segmented into 20-minute slices according to its provided partitioning scheme.

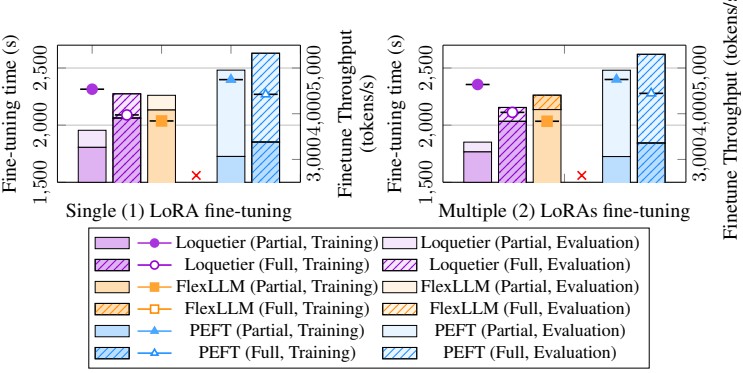

Figure 3: Comparison of the performance of Loquetier, FlexLLM and PEFT in fine-tuning tasks. The meanings of Partial and Full are the same as in Figure 2. × indicates that the results were not obtained: FlexLLM does not support backward propagation computations for modules other than up, gate and down. PEFT can only finetune one LoRA adapter at a time, so its time cost is cumulative.

**Hardware**. We test the inference-only tasks on a server with NVIDIA A6000 48G GPUs. We test the fine-tuning-only tasks and the unified fine-tuning and inference tasks on servers with NVIDIA H800 80G GPUs. Each test process has at least 128G of host memory available.

**Metrics and Tasks**. The metrics used for the evaluation are listed in Appendix C. The inference-only tasks run inference tasks at different request arrival rates where inference needs to be as fast as possible to achieve the Service Level Objective (SLO). The fine-tuning-only tasks need to run the training task for a specified number of epochs to measure its efficiency in processing tokens. The unified fine-tuning and inference tasks need to run the training task along with the inference task, weighing the efficiency of fine-tuning tokens against the SLO of the inference request. The real-world workload simulation experiment samples six time periods from the BurstGPT dataset, comprising one low-load, two medium-load, and three high-load intervals. Each slice was categorized into one of three load tiers based on its request rate: low-load periods for average RPS $< 1$; medium-load for 1~1.75; and high-load for $> 1.75$ (which may include transient spikes exceeding RPS 10). Here, RPS denotes requests per second. The detailed configurations can be viewed in Appendix D.

## 4.2 Evaluation results

**Model Loading**. The loading speed and additional loading storage overhead are shown in Table 2. Compared to PEFT, Loquetier requires creating Virtualized Modules and applying scaling to each LoRA linear when loading LoRA models, resulting in a slight slowdown in this part of the loading. FlexLLM needs to transform and cache the model weights, which leads to a significant additional storage footprint. Even with cached transformed models, FlexLLM's loading is still very slow due to the need to read small weight files.

**Inference**. Figure 2 shows the test results of the inference task. Loquetier maintains the highest SLO attainment at different request arrival rates. As the request rate increases, Loquetier's decoding speed gradually increases. Until at 3 RPS, the decoding speed no longer increases, indicating that the GPU memory access bottleneck has been hit. As the request rate continues to increase, some requests begin to fail to reach their SLOs due to the inability to achieve faster inference on the current GPU.

FlexLLM's maximum decoding speed is lower than Loquetier's, causing its SLO attainment rate to start dropping earlier and fall off a cliff at higher request arrival rates. In addition, in conjunction with the findings in Section 4.2, FlexLLM's lazy loading mechanism prevents it from handling some of the earliest arriving requests under SLO. Forcing early loading of the model weights improves SLO attainment to or near 100% at 1-2 RPS, but the improvement is very limited for higher request arrival rates due to the limitations of its highest decoding speed. FlexLLM is unable to apply LoRA to all 7 modules, causing it to fail under the corresponding experiments. When loading multiple LoRA models, FlexLLM is trapped in a dead loop for more than 10 minutes, missing SLOs for all requests. After a longer wait, FlexLLM cannot get out of the trap still, and therefore is marked as a failure.

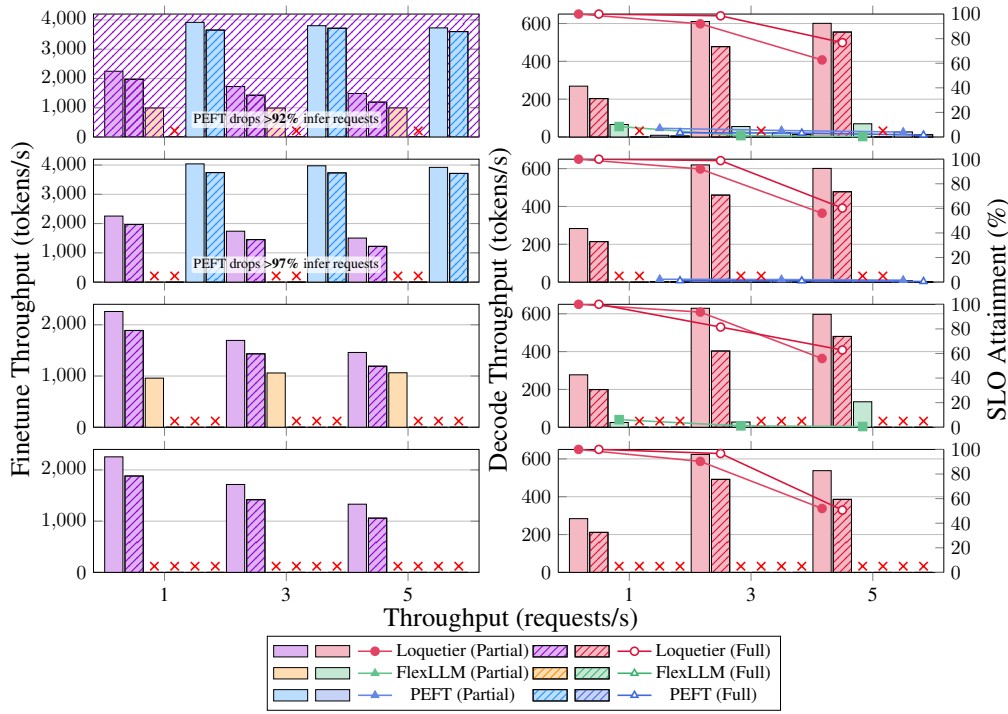

Figure 4: Comparison of the performance of Loquetier and PEFT in unified tasks. The 4 subplots correspond respectively to single-finetune & single-infer, single-finetune & multi-infer, multi-finetune & single-infer, and multi-finetune & multi-infer. The meanings of Partial and Full are the same as in Figure 2. × indicates that the results were not obtained: FlexLLM and PEFT can only finetune 1 LoRA at a time due to GPU memory limitations, causing it to fail the multi-LoRA fine-tuning scenarios; FlexLLM only support 3 target modules as mentioned in previous figures.

Transformers' batch strategy of padding different inputs to the same length makes its GPU memory footprint greatly affected by the batch size, making it very easy to trigger the error "CUDA out of memory". The processing speed of PEFT is constrained by the batch size to avoid exceeding the GPU memory. In multiple LoRA inference tasks, PEFT can only apply LoRAs in a serial for different configurations of inputs, making the inference speed further degraded. PEFT's SLO attainment rate is unacceptable even under 1 RPS.

**Fine-tuning**. The test results for the fine-tuning task are shown in Figure 3. The shorter total training time for Loquetier is due to its faster evaluation. Loquetier's fine-tuning is slightly slower than that of PEFT, mainly because of the independent computational calls from the LoRA linears during backward propagation. FlexLLM encounters an unsupported operation error during its peft backward propagation, indicating its inability to complete the experiments. The results show that Loquetier leads to almost no loss of fine-tuning efficiency.

**Unified Fine-tuning and Inference**. We test a combination of different configurations of fine-tuning and inference tasks, and the results are shown in Figure 4. Loquetier is able to provide an average of about 40% fine-tuning efficiency over three different request arrival rates while maintaining similar SLOs as in the inference-only tasks. PEFT's inference efficiency is too low, resulting in over 90% of the inference tasks timing out before they even begin. PEFT's fine-tuning tasks only drop about 20% efficiency, but this is due to the fact that PEFT has almost no computational overhead on the inference tasks, allowing the vast majority of the computing resources to still be used by the fine-tuning tasks. FlexLLM fails to complete the fine-tuning task, so it is not available for unified tests.

**Mutable Capacity Allocation Simulation**. In order to evaluate performance facing dynamic loads in real-world scenarios, we design an inference subtask with dynamic input throughput. As shown in Figure 5, Loquetier is able to adaptively adjust the efficiency of both fine-tuning and inference tasks under dynamic loads in the mutable unified task. The fine-tuning task makes concessions for

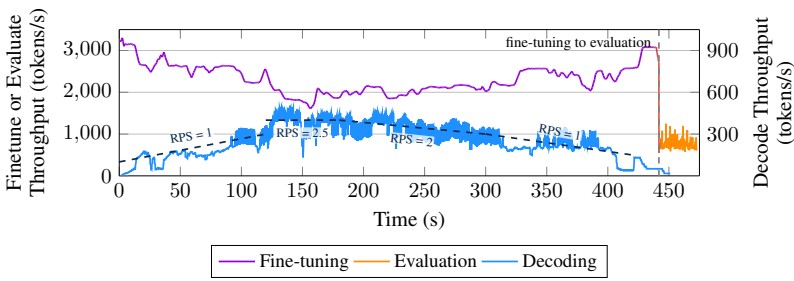

Figure 5: Performance of Loquetier under dynamic load in unified task.

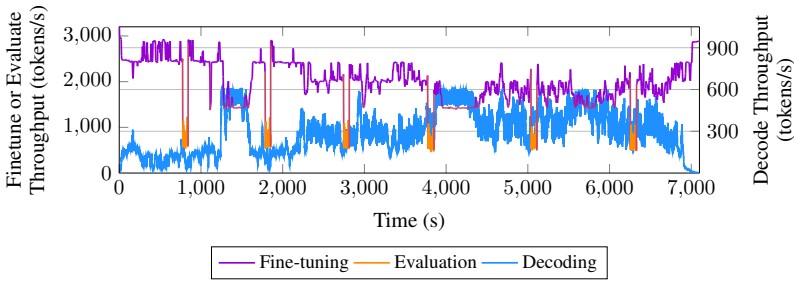

Figure 6: Performance of Loquetier under simulated real-world load in unified task.

the inference task to ensure the quality of service when request throughput increases, and adjusts back the efficiency by itself when throughput decreases.

**Simulated Real-world Workload**. To better simulate real-world serving conditions, we construct a composite workload using slices from the BurstGPT dataset, as mentioned in Section 4.1. Each slice contains request arrival times, input lengths, and output lengths. In our simulation, we fully utilize the request arrival times and reference the input length data. For an overview of sampling data and preference adjustments, see Appendix D.6. As shown in Figure 6, Loquetier demonstrates strong adaptability to real-world workloads, aligning closely with trends evaluated in the earlier simulation experiments. The final SLO for the entire experiment reaches 92.37%. All requests that failed to meet service metrics occur during transient workload spikes under high-load conditions (RPS > 5), which exceeded the load capacity of the hardware. In all other periods, Loquetier consistently achieves the defined SLO.

## 5   Conclusion

We present Loquetier, a virtualized multi-LoRA framework that runs fine-tuning and inference tasks uniformly. Loquetier performs well in the inference task, with an SLO attainment $20.8\times$ higher than PEFT, and up to $3.0\times$ that of FlexLLM at high request arrival rates, maintaining comparable efficiency in the fine-tuning task. In the unified task, inference efficiency is maintained as much as possible with an SLO attainment $46.4\times$ higher than PEFT, well balancing the performance of fine-tuning and inference tasks.

## Acknowledgments and Disclosure of Funding

We are thankful to the anonymous reviewers for their helpful comments. This work is supported by Jiangsu Science and Technology Major Project (#BG2025005), the National Natural Science Foundation of China (Grants #62172199), the Collaborative Innovation Center of Novel Software Technology and Industrialization, and Jiangsu Wukong Intelligent Computing Digital Technology. Jingwei Xu (jingweix@nju.edu.cn) is the corresponding author.

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

## A  Limitations

Loquetier can be improved by combination with other training and fine-tuning optimization methods, and by applying other cluster management systems to provide clusterized services. We plan to provide a backward propagation kernel operating in concert with the SMLM kernel to accelerate fine-tuning.

Co-operating with forward propagation computations during backward propagation can go some way to balancing the bottleneck of GPU memory bandwidth and computational resources for fine-tuning and inference tasks, and Loquetier can be further improved in this regard. We also consider providing support for other LoRA-like PEFT methods.

## B  Solutions of FlexLLM Backward Procedure Issues

Several cases of operation remain unimplemented in FlexLLM's gradient computation logic, including OP_GELU, OP_RELU, OP_SIGMOID, OP_TANH, and OP_ELU. We noticed that their repository contained forward and backward kernels related to these operations, but they had never applied these backward kernels to their computation flow, resulting in its inability to perform fine-tuning tasks. We ultimately built a runnable version by instantiating these kernels at the missing locations. This fix was based on our understanding of their framework, and we did not implement any additional computational steps. The correction will not result in a performance degradation of FlexLLM.

## C  Experimental Metrics

We use the following experimental metircs:

- **Service Level Objective (SLO)**: Measuring service satisfaction at a level.
- **SLO Attainment**: Percentage of all requests reaching the given SLO.
- **Request Throughput**: Throughput of incoming inference requests. Measured as request per second (RPS).
- **Decode Throughput**: Throughput of inference requests in decoding. Measured as decode token per second (DTPS).
- **Finetune Throughput**: Throughput of fine-tuning requests in training forward. Measured as finetune token per second (FTPS).
- **Evaluate Throughput**: Throughput of fine-tuning requests in evaluation forward. Measured as evaluate token per second (ETPS).

## D  Experimental Settings

### D.1  SLO

For all inference requests in the experiments, we use the following targets in Figure 3 as SLO.

Since PEFT performs batch generation with padding, resulting in additional computational delays in prefilling and decoding, we do not require more decoding aspects for PEFT.

Table 3: SLO settings.

| Framework or System | Max Waiting Time (s) | Mean Decoding Latency (ms) | Max Decoding Latency (ms) |
|---|---|---|---|
| Loquetier | 6 | 200 | 1,000 |
| PEFT | 6 | - | - |
| FlexLLM | 6 | 200 | 1,000 |

Table 4: Inference-only tasks configurations. The number of requests in multiple (4) LoRAs is expressed as the total number / number of one LoRA.

| Throughput (RPS) | Single (1) LoRA | | Multiple (4) LoRAs | |
|---|---|---|---|---|
| | Requests | Max New Tokens | Requests | Max New Tokens |
| 1 | 800 | 400 | 800 / 200 | 400 |
| 2 | 1,600 | 400 | 1,600 / 400 | 400 |
| 3 | 2,400 | 400 | 2,400 / 600 | 400 |
| 4 | 3,200 | 300 | 3,200 / 800 | 300 |
| 5 | 4,000 | 200 | 4,000 / 500 | 200 |

## D.2 Inference

We use the following configurations in Figure 4 to test the inference-only tasks.

Note that the maximum tokens supported by FlexLLM is 1024, so all inference requests for FlexLLM do not exceed this limit.

## D.3 Fine-tuning

We use the following configurations in Figure 5 to test the fine-tuning-only tasks.

Table 5: Fine-tuning-only tasks configurations.

| Configurations | Single (1) LoRA | Multiple (2) LoRAs |
|---|---|---|
| LoRA Config | | |
|     r (rank) | 8 | 8 |
|     lora_alpha | 16 | 16 |
|     lora_dropout | 0.05 | 0.05 |
|     bias | none | none |
|     task_type | CAUSAL_LM | CAUSAL_LM |
|     init_lora_weights | gaussian | gaussian |
| Training Args | | |
|     per_device_train_batch_size | 2 | 1 |
|     per_device_eval_batch_size | 2 | 1 |
|     num_train_epochs | 4 | 4 |
|     eval_strategy | epoch | epoch |
|     logging_strategy | steps | steps |
|     logging_steps | 100 | 100 |
|     save_strategy | epoch | epoch |
|     learning_rate | 2e-5 | 2e-5 |
|     gradient_accumulation_steps | 4 | 4 |
|     report_to | none | none |

## D.4 Unified fine-tuning and inference

We use the following configurations in Figure 6 to test the unified tasks.

Note that the maximum tokens supported by FlexLLM is 1024, so all inference requests for FlexLLM do not exceed this limit.

## D.5 Mutable capacity allocation simulation

We use the following configurations in Figure 7 for inference requests and the single LoRA fine-tuning configurations in Figure 6 for fine-tuning requests to test the mutable unified tasks.

Table 6: Unified tasks configurations.

| Throughput (RPS) | Single (1) LoRA | | Multiple (4) LoRAs | |
|---|---|---|---|---|
| | Requests | Max New Tokens | Requests | Max New Tokens |
| 1 | 600 | 400 | 600 / 150 | 400 |
| 2 | 1,200 | 400 | 1,200 / 300 | 400 |
| 3 | 1,800 | 400 | 1,800 / 450 | 400 |
| 4 | 2,400 | 300 | 2,400 / 600 | 300 |
| 5 | 3,000 | 200 | 3,000 / 750 | 200 |

| Configurations | Single (1) LoRA | Multiple (2) LoRAs |
|---|---|---|
| LoRA Config | | |
|     r (rank) | 8 | 8 |
|     lora_alpha | 16 | 16 |
|     lora_dropout | 0.05 | 0.05 |
|     bias | none | none |
|     task_type | CAUSAL_LM | CAUSAL_LM |
|     init_lora_weights | gaussian | gaussian |
| Training Args | | |
|     per_device_train_batch_size | 2 | 1 |
|     per_device_eval_batch_size | 2 | 1 |
|     num_train_epochs | 1 | 1 |
|     eval_strategy | epoch | epoch |
|     logging_strategy | steps | steps |
|     logging_steps | 100 | 100 |
|     save_strategy | epoch | epoch |
|     learning_rate | 2e-5 | 2e-5 |
|     gradient_accumulation_steps | 4 | 4 |
|     report_to | none | none |

Table 7: Mutable unified tasks configurations.

| | Multiple (4) LoRAs | | | | |
|---|---|---|---|---|---|
| Index | LoRA Index | Requests | Throughput (RPS) | Start at (s) | Duration (s) |
| 1 | 0 | 120 | 1 | 0 | 120 |
| 2 | 1 | 150 | 2.5 | 120 | 60 |
| 3 | 2 | 240 | 2 | 180 | 120 |
| 4 | 3 | 120 | 1 | 300 | 120 |

## D.6 Simulated real-world workload

We use the following time periods in Figure 8 to test (the inference task of) the simulated real-world workload. The fine-tuning task use the same configuration as mutable capacity allocation simulation.

Based on our analysis of the BurstGPT dataset, less than one-third of the time periods correspond to high-load scenarios, with the majority being low-load periods. Given the lower challenge of low-load periods, we adopt the configuration above to ensure representative coverage. Among the selected workloads, high-load periods include several minutes where the RPS exceeded 5, with a peak RPS of 11.

## E Detailed Information Related to S-LoRA in the Experiment

S-LoRA does not support the LLaMA 3 series models, and its repository has been archived. This limitation arises from the Group Query Attention (GQA) architecture used in LLaMA 3, where the shapes of K and V differ from those of Q and O. Consequently, the shape of the weight matrix B in the LoRA linear layers for K and V also differs from Q and O. Current S-LoRA requires all LoRA

Table 8: Time periods configurations. Peak RPS refers to the highest RPS within a 2-second interval.

| Time Period | Requests | Mean RPS | Peak RPS |
|---|---|---|---|
| Day 29, 13:00 ~13:20 | 676 | 0.563 | 1.5 |
| Day 29, 15:00 ~15:20 | 2,145 | 1.788 | 11.5 |
| Day 29, 16:00 ~16:20 | 1,465 | 1.226 | 7 |
| Day 33, 13:40 ~14:00 | 2,823 | 2.354 | 10 |
| Day 33, 11:40 ~12:00 | 2,360 | 1.966 | 12 |
| Day 33, 11:00 ~11:20 | 1,856 | 1.547 | 10.5 |

weights within the same layer to be concatenated at runtime. However, due to the shape discrepancies mentioned above, this concatenation operation fails. As a workaround, we replicate K and V weights in advance during model initialization to enable S-LoRA to start properly.

"Partial" in Figure 2 means that only 4 modules are enabled for S-LoRA including q, k, v, and o, as S-LoRA supports applying LoRA only on these 4 linear layers, and does not support the up, gate, and down layers within the MLP. Therefore, its runtime efficiency resembles the Partial scenario described in our paper, where only three linear layers (up, gate, down) are targeted.

In experiments, we observed instability in the S-LoRA kernel, which frequently produced incorrect outputs leading to NaN or Inf values. These errors propagate quickly, causing model generation failures. At this time, we did not modify the kernel. Our preliminary analysis suggests this may be due to missing synchronization mechanisms in some computational steps. (Note that this is an initial observation and may not be definitive.)

Due to these issues, S-LoRA struggled to complete all inference requests in our scenarios, as it frequently outputs the eos token directly, making SLO appear better than it actually should be.

