# OpenReview forum: "Loquetier: A Virtualized Multi-LoRA Framework for Unified LLM Fine-tuning and Serving"
_NeurIPS.cc/2025/Conference — NeurIPS 2025 poster_

### Official Review · Reviewer_VwoD · 2025-06-01

**Clarity:** 2
**Significance:** 2
**Originality:** 3
**Rating:** 3
**Confidence:** 4

**Summary:**

The paper presents Loquetier, a unified framework that integrates low-rank adaptation (LoRA) fine-tuning and inference for large language models, addressing the scalability challenges in parameter‐efficient fine-tuning techniques. The authors motivate their work by highlighting the increasing computational costs of full-model training and the need for dynamic, runtime-adaptable systems that support multiple fine-tuning configurations concurrently. The proposed solution leverages a Virtualized Module to isolate model modifications and an optimized computation flow with the Segmented Multi-LoRA Multiplication (SMLM) kernel to merge fine-tuning and inference processes. Experiments show that Loquetier outperforms competing baselines in both performance and flexibility under various task settings.

**Questions:**

See weaknesses.

**Ethical Concerns:**

["NO or VERY MINOR ethics concerns only"]

**Final Justification:**

Thanks for the rebuttal. Some of my concerns are addressed. I'd like to increase my score.

**Limitations:**

This work has discussed its limitations.

**Quality:**

2

**Strengths And Weaknesses:**

Strengths:

1. The framework introduces a SMLM kernel that unifies fine-tuning and inference, enabling mixed-task execution with a shared forward propagation path. The kernel design minimizes kernel invocation overhead and streamlines memory management. The approach effectively balances the computational needs of dynamic workloads in large language model serving.

2. The Virtualized Module mechanism in the framework provides clean isolation of LoRA modifications, allowing for independent and concurrent execution of multiple adapter configurations. The design allows for instance-level migration without incurring memory duplication or requiring kernel restarts. The modular abstraction enhances flexibility and supports a wide variety of parameter-efficient fine-tuning methods.

3. Experiment results across inference-only, fine-tuning-only, and unified tasks show the performance gains of Loquetier. The evaluation demonstrates significant improvements in decoding throughput and service level objective attainment compared to established baselines.

4. The framework is compatible with existing LLM inference optimizations, demonstrating interoperability with kernels such as FlashInfer and strategies like dynamic caching. The ability to leverage state-of-the-art inference techniques enhances its practical applicability. The solution shows versatility by supporting various PEFT methods without requiring extensive changes to the base model architecture.


Weaknesses:

1. The slight slowdown observed during model loading due to the creation of Virtualized Modules adds additional latency and may affect real-time system scalability. The trade-off between isolation and initialization efficiency is not fully explored, leaving some uncertainty about performance in latency-critical environments.

2. The framework is primarily tested on LoRA-based PEFT techniques and may not generalize seamlessly to other fine-tuning methods or modifications outside of the LoRA family. The experimental evaluation does not include a variety of alternative PEFT methods, which limits insights into broader applicability.

3. The experimental setup only includes the LLaMA-8B model and the shareGPT, alpaca and GSM8k datasets. However, many real-world datasets that require finetuning and personalization should be considered, which is the core motivation of providing large-scale finetuning services.

4. The writing should be largely improved. Considering that authors propose a new setting, there should be a clear introduction of what are inputs and outputs. How do users use this system and what their dataflow? While the motivation of providing finetuning and serving of PEFT methods is clear, the motivation of the framework design is unclear.

5. While this work provides a new kernel optimization and a Virtualized Module, from the system perspective, there still lacks a clear optimization of how the memory and computation are optimized. This is partly because that authors do not clearly describe the scenario. It is recommended to add some sections to clearly describe the scenarios, using some timeline figures to convey challenges and opportunities of system improvement. Besides, only the code implementation is a weak contribution from the system perspective.

---

> ### Author Rebuttal · Authors · 2025-07-31
>
> Thank you for your detailed review of our work. Before responding to the weaknesses and questions you raised, please allow us to once again clarify our research background, scenario definition, and the experimental metrics used to measure results.
>
> # Background, Scenario Definition, and Key Experimental Metrics
>
> First, please allow us to reiterate our research background, scenario definition, and the experimental metrics used to measure results.
>
> Loquetier is a framework for unified fine-tuning and serving of multiple LoRAs. As mentioned in the Introduction, the research background is that LoRA and its related efficient fine-tuning techniques have demonstrated excellent performance across various specialized tasks and show significant potential in many tasks, such as personalization and user preference-related tasks, leading to their increasingly widespread adoption and promising future applications. However, the hardware resources available for LLM training, fine-tuning, and serving remain extremely limited. This leads to our application scenario: how to improve the efficiency of multi-LoRA fine-tuning and inference tasks under constrained hardware conditions.
>
> Specifically, most existing system frameworks require a hard switch when handling different tasks, meaning they must first stop the current task before starting another. Such switches consume significant time and require saving fine-tuning states and migrating inference requests, which occupy substantial bandwidth and storage space. During downtime, the occupied computational resources cannot be utilized, leading to a significant decline in the system's hardware resource utilization efficiency.
>
> Unifying fine-tuning and inference of multiple LoRAs upon one base model can address the aforementioned issues and improve the efficiency of multi-LoRA fine-tuning and inference tasks. Integrating the two distinct tasks is a practical approach, as the only difference between them lies in the gradient calculation and parameter updates during the backpropagation process. Regardless of the task type, sharing the same base model can significantly reduce memory requirements.
>
> According to the description in the Experiments section, we measure the efficiency of these two tasks as follows, and list the meanings of all experimental metrics in Appendix B:
>
>  - The efficiency metric for the fine-tuning task is the throughput of fine-tuning and evaluation, i.e., the number of tokens fine-tuned and evaluated per unit of time, which is equivalent to the time required to fine-tune with the same data and configuration.
>
>  - The efficiency metric for the inference task is inference throughput, which is the number of tokens inferred per unit of time. Additionally, serving metrics targets are used to define the availability of the service system, so the secondary metric is the service level objective attainment (rate). Appendix C clarifies the service metric settings, including the maximum request wait time, maximum decoding latency per token, and average decoding latency. By calculating whether each request meets these strict serving metrics, our experimental results have sufficient practical significance.
>
> # Latency Impact Caused by Virtualized Module
>
> Due to the objective limitations on the number of LoRAs that can be loaded on each GPU, in practical use, we can pre-create a sufficient number of virtual base models for the maximum number of LoRAs that can be accommodated, along with an equivalent number of empty LoRAs, and perform data copying when actually loading LoRAs. Even if these virtual modules need to be dynamically created, one or two pre-prepared virtual models can always be reserved. In this case, the latency introduced by Virtualized Module can be resolved during the framework startup phase, so this latency does not affect performance and efficiency in latency-critical environments.
>
> # Extension to other PEFT methods
>
> As stated in the paper and the first section "Background, Scenario Definition, and Key Experimental Metrics" in our response, Loquetier is a framework for unified fine-tuning and inference task design in multi-LoRA scenarios, rather than overall optimization of various PEFT methods. We use a Virtualized Module mechanism to support the loading of other PEFT models. We also support optimization work for other PEFT methods. For example, if a study has designed an efficient operator for a certain method, we only need to introduce this operator into Loquetier to achieve optimization of this method.
>
> # Models and Datasets
>
> We did not propose improvements to the effectiveness for fine-tuning and inference methods, but rather optimizations to the computational processes for multi-LoRA fine-tuning and inference tasks. So there is no need for validating the effectiveness of fine-tuning on vast downstream tasks with F-1, rouge-L, etc. The current dataset is sufficient to evaluate all the experimental metrics mentioned earlier.
>
> In this paper, we made improvements at the kernel level. The industry typically evaluates computational quality through the following two parts of validation, and based on our validation results, Loquetier does not affect the quality of computational results.
>
>   1. Verifying whether kernel computational outputs align with standard results: We performed 10,000 kernel operations using random inputs, and the results indicate that no errors were generated in kernel computational results at float16 precision.
>   2. Verifying whether the loss converges normally during fine-tuning: Based on data generated during the experiment, the loss converges consistently with the PEFT baseline method.
>
> Regarding the dataset used for inference tasks, we applied the same data input fairly to all test baselines. The data used reflects the highest throughput of our framework, and this result is not influenced by the dataset, verifying that the inference task has achieved its objectives. Additionally, we designed a unified task for fine-tuning and inference with dynamic input scenarios within hardware constraints (Section 4.2 - Unified Fine-tuning and Inference). Experimental results show that Loquetier can handle any dynamic input changes under hardware limitations.
>
> # Framework Design and Motivation
>
> As mentioned in the paper and the first section "Background, Scenario Definition, and Key Experimental Metrics" in the response, there is a lack of work on multi-LoRA fine-tuning and inference in the context we describe. Therefore, we designed a unified framework for fine-tuning and inference to significantly improve the execution efficiency of these tasks and hardware utilization in scenarios with limited hardware resources, while maintaining good compatibility with other PEFT methods and related optimization work.
>
> Loquetier does not modify user input or output, which is consistent with other related work. Specifically, each request only needs to provide its text content or token sequence, along with the specified LoRA (LoRA ID). The framework will return the generated text content or token sequence sequentially. For fine-tuning requests, we save the obtained model in the specified file path, and immediate fine-tuning results can be obtained by accessing the originally created LoRA model. The data flow is illustrated in Figure 1 (architecture diagram) of the paper. Due to NeurIPS restrictions on images and external links in replies, we cannot reproduce the image here; please refer to the submitted paper.
>
> # Scenario and Contribution
>
> As noted in the paper and the first section "Background, Scenario Definition, and Key Experimental Metrics" in our response, our scenario involves fine-tuning and inference tasks using multiple LoRA models under limited hardware resource constraints, with the goal of improving the efficiency of these tasks in such scenarios.
>
> Our memory optimization is achieved through Virtualized Module, a mechanism that enables different LoRA or PEFT models to share the same base model. By loading only one base model, we significantly reduce GPU memory usage in multi-LoRA scenarios. The specific virtualized model architecture is detailed in Figure 1 (architecture diagram) of the paper. Our computational optimization synchronizes computations by loading different data into a single kernel call for input data and various LoRA weights, thereby efficiently utilizing hardware resources to accelerate the computational process.
>
> Our work includes over 1,000 lines of CUDA kernel and its related code, nearly 4,000 (3,893) lines of code supporting kernel design, fine-tuning, inference workflows, and model adaptation, as well as over 1,000 lines of code supporting PyTorch native training. We also made modifications to the kernel calls in FlashInfer to support Llama 3's RoPE and other settings.
>
> # Additional Supplement
>
> Please allow us to update the performance of FlexLLM in multiple LoRA inference scenarios. We conducted a smaller-scale test, and the results showed that FlexLLM ran for several hours while repeatedly loading LoRA in a loop for a test that should not have lasted more than 5 minutes—it indeed did not get stuck in an infinite loop. We have not studied its underlying implementation in detail, so we will not speculate on the cause of this phenomenon.

---

> > ### Comment · Reviewer_VwoD · 2025-08-01
> >
> > Thanks for the rebuttal. Some of my concerns are addressed. I'd like to increase my score.

---

> > > ### Author Response · Authors · 2025-08-04
> > >
> > > We appreciate and respect your review and your acknowledgment of our response. Thank you for sharing your concerns, opinions, and suggestions with us based on your professional knowledge.

---

> > > ### Author Response · Authors · 2025-08-07
> > >
> > > Based on the reviewers' suggestions, we have completed the main part of the FlexLLM experiment, and we would like to share the results with you.
> > >
> > > In our paper, we mentioned that FlexLLM cannot run fine-tuning tasks in their official Docker environment, as it is missing at least one operator in the backward computation process. After investigation, we confirmed that the missing operator is OP_SCALAR_TRUE_DIV. Additionally, we noted that five other operators from the same source (the header file element_unary.h) are also missing: OP_GELU, OP_RELU, OP_SIGMOID, OP_TANH, and OP_ELU.
> > >
> > > Before submitting the paper, we spent a considerable amount of time attempting to build their framework, but our efforts ended in failure. Therefore, we only included results of FlexLLM in certain scenarios in the paper. Over the past few days, we attempted to build dozens of commits dating back to Feb 2024 (FlexLLM paper v1 published). encountering compilation and runtime issues such as source code errors, dependency errors, segmentation faults, and PEFT model incompatibility. In all these versions, instances of the aforementioned operators were never included in the backward computation process.
> > >
> > > We noticed that the FlexLLM repository contained forward and backward kernels related to element_unary, but they had never applied these backward kernels to fused.cu, resulting in its inability to perform fine-tuning tasks. **We ultimately built a runnable version by instantiating these kernels at the missing locations. This fix was based on our understanding of the FlexLLM framework, and we did not implement any additional computational steps. The correction will not result in a performance degradation of FlexLLM.** As mentioned earlier, we want you to know that we had previously made every effort to reproduce its functionality, and the reason for the failure to reproduce was not due to us.
> > >
> > > The experiment comparison results for fine-tuning and unified tasks including FlexLLM are as follows; all experimental settings (including metric settings) are consistent with those in the paper; other data are taken from the experimental results in the paper:
> > >
> > > ## Finetuning
> > >
> > > | | FlexLLM | | | | Loquetier | | | | PEFT | | | |
> > > | - | - | - | - | - | - | - | - | - | - | - | - | - |
> > > | Setup | Total Time (s) | FTime (s) | ETime (s) | FTP (tps) | Total Time | FTime | ETime | FTP | Total Time | FTime | ETime | FTP |
> > > | S | 2261.28 | 2132.35 | 128.93 | 3841.77 | 1955.14 | 1805.75 | 149.39 | 4536.62 | 2481.63 | 1726.17 | 755.46 | 4745.76 |
> > > | M | 2262.46 | 2135.99 | 126.47 | 3835.22 | 1851.45 | 1765.75 | 85.7 | 4639.4 | 2480.78 | 1724.94 | 755.84 | 4749.16 |
> > >
> > > ## Unified
> > >
> > > | | | FlexLLM | | | Loquetier | | | PEFT | | |
> > > | - | - | - | - | - | - | - | - | - | - | - |
> > > | Setup | RPS | FTP (tps) | DTP (tps) | SLO (%) |  FTP | DTP | SLO | FTP | DTP | SLO |
> > > | S | 1 | 990.58 | 66.13 | 8.5 | 2246.14 | 268.9 | 100 | 3916.88 | 9.09 | 7.06 |
> > > | S | 3 | 988.42 | 55.32 | 1.11 | 1724.66 | 610.54 | 92.06 | 3802.13 | 21.96 | 5.29 |
> > > | S | 5 | 994 | 69.88 | 0.37 | 1487.4 | 600.87 | 62.67 | 3727.26 | 26.95 | 4.01 |
> > > | M | 1 | 958.38 | 25 | 6.17 | 2256.49 | 283.37 | 100 | 4039.2 | 3.18 | 2.34 |
> > > | M | 3 | 1059.39 | 27.63 | 1.11 | 1739.41 | 619.66 | 92 | 3974.24 | 8.12 | 2.11 |
> > > | M | 5 | 1063.36 | 134.72 | 0.57 | 1505.21 | 601.2 | 56.07 | 3920.17 | 7.52 | 1.6 |
> > >
> > > * All experiments used Partial (which means 3 target modules).
> > > * In Setup, S/M denotes single/multiple LoRA; S/M in unified tasks denotes the number of LoRA in inference tasks, while fine-tuning tasks are all single LoRA.
> > > * tps denotes tokens per second, FTime/ETime denotes Fine-tuning/Evaluation Time, FTP/DTP denotes Fine-tuning/Decoding throughput, SLO denotes Service Level Objective (Attainment).
> > > * Configurations that appear in the paper but are not listed in the table above are not supported by FlexLLM. The issues with target modules have been identified in the paper, and the relevant experiments in the unified task can only perform single LoRA fine-tuning because FlexLLM cannot perform multi-LoRA fine-tuning.
> > > * FlexLLM will only execute one fine-tuning task at a time, even if two fine-tuning requests are submitted simultaneously; the fine-tuning instances of FlexLLM consume far more than 40G of GPU memory (H800 has 80G). This means that in our scenarios, similar to PEFT, FlexLLM can only run multiple fine-tuning requests sequentially. The data in the table is the result of sequential runs.
> > >
> > > The empirical results above are consistent with our previous findings, namely that Loquetier outperforms FlexLLM in these scenarios.
> > >
> > > Based on previous proxy evaluations and the results above, we are even more confident that Loquetier outperforms the S-LoRA+PEFT baseline. We will complete the remaining experiments as soon as possible and share the results.

---

### Official Review · Reviewer_tvxa · 2025-06-27

**Clarity:** 3
**Significance:** 3
**Originality:** 3
**Rating:** 4
**Confidence:** 2

**Summary:**

This work presents Loquetier, a virtualized multi-LoRA framework that enables unified LoRA fine-tuning and serving. Specifically, Loquetier consists of two main components: (1) a Virtualized Module that isolates model modifications to support multiple PEFT methods within a shared base model architecture, and (2) an optimized computation flow and kernel design that merges fine-tuning and inference paths in forward propagation, enabling efficient batching and minimizing kernel invocation overhead. Experimental results demonstrate its effectiveness.

**Questions:**

none

**Ethical Concerns:**

["NO or VERY MINOR ethics concerns only"]

**Final Justification:**

The author's response has successfully resolved some of my concerns. I tend to keep my score.

**Limitations:**

yes

**Quality:**

3

**Strengths And Weaknesses:**

Strength:

1. This work investigates an important problem: how to unify fine-tuning and inference processes for LoRA-based models?

2. The paper is well written and clearly structured. Also, the proposed method is effective compared to baselines.

3. The authors provide code for reproduction.

Weakness:

This is **not** my area of expertise (I will also give **low confidence** for my rating), so I may have missed some potential weaknesses. One potential concern is that the evaluation only includes FlexLLM as a baseline. However, the "Related Work" section mentions several other methods targeting efficient multi-LoRA inference. Including comparisons with them may be necessary.

Besides, the experiments are limited to Llama3-8B. Whether the findings can be generalized to larger models remains unverified.

Finally, it would be better to evaluate the method on more challenging tasks beyond alpaca and GSM8K.

---

> ### Author Rebuttal · Authors · 2025-07-31
>
> Thank you for your detailed review of our work. Our responses to the weaknesses you raised are as follows:
>
> # Baselines
>
> We have received valuable suggestions from the reviewers regarding the evaluation. We would like to provide additional explanations and the derived results to address the issues raised.
>
> For S-LoRA, FlexLLM have compared with it. S-LoRA is only optimized for inference tasks and performs worse than FlexLLM in fine-tuning and inference tasks; FlexLLM's experiments include the Peft baseline for fine-tuning tasks, and our results in similar experimental settings are better than the equivalent fine-tuning performance of Peft in the S-LoRA+Peft baseline in FlexLLM's experiments, while the performance of the Peft is better than that of FlexLLM in several experimental environments (FlexLLM has not tested a direct comparison with Peft when only performing fine-tuning tasks; based on the result in their paper and the number of GPUs used, we can infer that Peft's fine-tuning efficiency is higher than that of FlexLLM under the same hardware resources). Therefore, based on the proxy evaluation method, we can also confirm that our performance is better than that of FlexLLM and S-LoRA. Additionally, S-LoRA significantly outperforms vLLM in its multi-LoRA inference experiments, and its experimental environment is comparable to ours. Furthermore, S-LoRA does not support newer models, including Llama 3. For these reasons, we did not design experiments comparing S-LoRA and vLLM.
>
> As mentioned in the Introduction, there are very few training-inference integrated frameworks that support multi-LoRA. FlexLLM may be the only proper baseline. The other methods mentioned in the Related work are mostly general optimizations for base models or further improvements to the LoRA method itself, and they are not suitable for evaluation as test baselines; our work is also compatible with these methods.
>
> For the performance of FlexLLM in multiple LoRA inference scenarios. We conducted a smaller-scale test, and the results showed that FlexLLM ran for several hours while repeatedly loading LoRA in a loop for a test that should not have lasted more than 5 minutes—it indeed did not get stuck in an infinite loop. We have not studied its underlying implementation in detail, so we will not speculate on the cause of this phenomenon.
>
> # Models
>
> Since we are not an enterprise-level organization and are constrained by hardware resources, it is challenging to conduct experiments with larger models. We are willing to analyze the expected results of larger models on multiple GPUs from a theoretical perspective.
>
> The LoRA method itself does not conflict with common parallel strategies, such as pipeline parallelism or tensor parallelism. Therefore, from a technical standpoint, the fine-tuning and inference of LoRA models can utilize these parallel strategies to load larger models on multiple GPUs. In terms of pipeline parallelism, data only needs to be transferred once between the two layers on different devices, so our work fully supports this parallel strategy. We also fully support parallel strategies like tensor parallelism that require data sharing between GPUs:
>
> The forward and backward calculations of LoRA can be expressed by the following formulas:
>
> $$V = X @ A,  Y = V @ B$$
>
> $$B' = V^T @ Y',  V' = Y' @ B^T, A' = X^T @ V', X' = V' @ A^T$$
>
> Data parallelism is typically split along the hidden_states dimension, so the weight matrix needs to be split and stored across multiple GPUs, with the requirement of an all-reduce or all-gather performed after each of the $V, Y, V',$ and $X'$ computations are complete. Loquetier's computation process can be split using all-reduce / all-gather, so we also support these parallelization strategies.
>
> This means we can run larger models in a multi-GPU environment without incurring any additional overhead compared to existing methods.
>
> One point we would like to highlight is that, based on the relevant research we have collected, there has been little targeted optimization for such parallel strategies when using LoRA, especially in fine-tuning tasks, as the LoRA computation itself already has no further room for parallel optimization. Furthermore, for strategies like tensor parallelism, as shown in the above formula, the communication overhead required for each LoRA linear layer to synchronize is  $2 \times s \times (r + h) \times n$, where $n$ is the number of GPUs involved in parallelism. Considering the number of LoRA layers and the dimension of hidden_size in large models, the communication overhead is extremely high compared to the benefits, so these parallel strategies is typically not used.
>
> # Datasets
>
> We did not propose improvements to the effectiveness for fine-tuning and inference methods, but rather optimizations to the computational processes for multi-LoRA fine-tuning and inference tasks. So there is no need for validating the effectiveness of fine-tuning on vast downstream tasks with F-1, rouge-L, etc. The current dataset is sufficient to evaluate all the experimental metrics mentioned earlier.
>
> In this paper, we made improvements at the kernel level. The industry typically evaluates computational quality through the following two parts of validation, and based on our validation results, Loquetier does not affect the quality of computational results.
>
>   1. Verifying whether kernel computational outputs align with standard results: We performed 10,000 kernel operations using random inputs, and the results indicate that no errors were generated in kernel computational results at float16 precision.
>
>   2. Verifying whether the loss converges normally during fine-tuning: Based on data generated during the experiment, the loss converges consistently with the PEFT baseline method.

---

> > ### Comment · Reviewer_tvxa · 2025-08-05
> > **Response to authors**
> >
> > Thanks for the clarification. I will maintain my score.

---

> > > ### Author Response · Authors · 2025-08-07
> > >
> > > We appreciate and respect your review and your acknowledgment of our response. Thank you for sharing your concerns, opinions, and suggestions with us based on your professional knowledge.
> > >
> > > In addition, we have completed the main part of the FlexLLM experiment and would like to share the results with you.
> > >
> > > In our paper, we mentioned that FlexLLM cannot run fine-tuning tasks in their official Docker environment, as it is missing at least one operator in the backward computation process. After investigation, we confirmed that the missing operator is OP_SCALAR_TRUE_DIV. Additionally, we noted that five other operators from the same source (the header file element_unary.h) are also missing: OP_GELU, OP_RELU, OP_SIGMOID, OP_TANH, and OP_ELU.
> > >
> > > Before submitting the paper, we spent a considerable amount of time attempting to build their framework, but our efforts ended in failure. Therefore, we only included results of FlexLLM in certain scenarios in the paper. Over the past few days, we attempted to build dozens of commits dating back to Feb 2024 (FlexLLM paper v1 published). encountering compilation and runtime issues such as source code errors, dependency errors, segmentation faults, and PEFT model incompatibility. In all these versions, instances of the aforementioned operators were never included in the backward computation process.
> > >
> > > We noticed that the FlexLLM repository contained forward and backward kernels related to element_unary, but they had never applied these backward kernels to fused.cu, resulting in its inability to perform fine-tuning tasks. **We ultimately built a runnable version by instantiating these kernels at the missing locations. This fix was based on our understanding of the FlexLLM framework, and we did not implement any additional computational steps. The correction will not result in a performance degradation of FlexLLM.** As mentioned earlier, we want you to know that we had previously made every effort to reproduce its functionality, and the reason for the failure to reproduce was not due to us.
> > >
> > > The experiment comparison results for fine-tuning and unified tasks including FlexLLM are as follows; all experimental settings (including metric settings) are consistent with those in the paper; other data are taken from the experimental results in the paper:
> > >
> > > ## Finetuning
> > >
> > > | | FlexLLM | | | | Loquetier | | | | PEFT | | | |
> > > | - | - | - | - | - | - | - | - | - | - | - | - | - |
> > > | Setup | Total Time (s) | FTime (s) | ETime (s) | FTP (tps) | Total Time | FTime | ETime | FTP | Total Time | FTime | ETime | FTP |
> > > | S | 2261.28 | 2132.35 | 128.93 | 3841.77 | 1955.14 | 1805.75 | 149.39 | 4536.62 | 2481.63 | 1726.17 | 755.46 | 4745.76 |
> > > | M | 2262.46 | 2135.99 | 126.47 | 3835.22 | 1851.45 | 1765.75 | 85.7 | 4639.4 | 2480.78 | 1724.94 | 755.84 | 4749.16 |
> > >
> > > ## Unified
> > >
> > > | | | FlexLLM | | | Loquetier | | | PEFT | | |
> > > | - | - | - | - | - | - | - | - | - | - | - |
> > > | Setup | RPS | FTP (tps) | DTP (tps) | SLO (%) |  FTP | DTP | SLO | FTP | DTP | SLO |
> > > | S | 1 | 990.58 | 66.13 | 8.5 | 2246.14 | 268.9 | 100 | 3916.88 | 9.09 | 7.06 |
> > > | S | 3 | 988.42 | 55.32 | 1.11 | 1724.66 | 610.54 | 92.06 | 3802.13 | 21.96 | 5.29 |
> > > | S | 5 | 994 | 69.88 | 0.37 | 1487.4 | 600.87 | 62.67 | 3727.26 | 26.95 | 4.01 |
> > > | M | 1 | 958.38 | 25 | 6.17 | 2256.49 | 283.37 | 100 | 4039.2 | 3.18 | 2.34 |
> > > | M | 3 | 1059.39 | 27.63 | 1.11 | 1739.41 | 619.66 | 92 | 3974.24 | 8.12 | 2.11 |
> > > | M | 5 | 1063.36 | 134.72 | 0.57 | 1505.21 | 601.2 | 56.07 | 3920.17 | 7.52 | 1.6 |
> > >
> > > * All experiments used Partial (which means 3 target modules).
> > > * In Setup, S/M denotes single/multiple LoRA; S/M in unified tasks denotes the number of LoRA in inference tasks, while fine-tuning tasks are all single LoRA.
> > > * tps denotes tokens per second, FTime/ETime denotes Fine-tuning/Evaluation Time, FTP/DTP denotes Fine-tuning/Decoding throughput, SLO denotes Service Level Objective (Attainment).
> > > * Configurations that appear in the paper but are not listed in the table above are not supported by FlexLLM. The issues with target modules have been identified in the paper, and the relevant experiments in the unified task can only perform single LoRA fine-tuning because FlexLLM cannot perform multi-LoRA fine-tuning.
> > > * FlexLLM will only execute one fine-tuning task at a time, even if two fine-tuning requests are submitted simultaneously; the fine-tuning instances of FlexLLM consume far more than 40G of GPU memory (H800 has 80G). This means that in our scenarios, similar to PEFT, FlexLLM can only run multiple fine-tuning requests sequentially. The data in the table is the result of sequential runs.
> > >
> > > The empirical results above are consistent with our previous findings, namely that Loquetier outperforms FlexLLM in these scenarios.
> > >
> > > Based on previous proxy evaluations and the results above, we are even more confident that Loquetier outperforms the S-LoRA+PEFT baseline. We will complete the remaining experiments as soon as possible and share the results.

---

### Official Review · Reviewer_WdRD · 2025-07-01

**Clarity:** 2
**Significance:** 3
**Originality:** 3
**Rating:** 4
**Confidence:** 4

**Summary:**

The paper presents Loquetier, a virtualized multi-LoRA framework that unifies fine-tuning and serving. It features virtualized modules that isolate model modifications, along with an optimized computation flow and kernel design that merges training and inference paths. The authors conduct an empirical study using three workloads and demonstrate that Loquetier consistently outperforms existing baselines.

**Questions:**

Please refer to the Weakness section. The paper contains several major issues, and due to their breadth and severity, it's difficult to distill them into a concise set of specific questions.

**Ethical Concerns:**

["NO or VERY MINOR ethics concerns only"]

**Final Justification:**

Thank you for the great effort during the rebuttal phase and the comprehensive clarification. All of my concerns are addressed. I'm raising my score to 4.

**Limitations:**

Yes.

That said, the listed limitations appear to be superficial and do not address the paper's deeper methodological or conceptual shortcomings.

**Quality:**

3

**Strengths And Weaknesses:**

## Strength

The paper introduces the Segmented Multi-LoRA Multiplication (SMLM) kernel and leverages it to unify fine-tuning and inference tasks with multiple LoRA adapters. This architectural direction is promising and relevant for scalable multi-task LLM deployment.

## Weakness

### Writing Quality

The paper is not well written, and suffers from a lack of clarity and structure. A few specific symptoms include:

**Unclear motivation:**

The motivation for unifying fine-tuning and inference with multi-LoRA is difficult to follow. On the surface, this work appears to be a natural extension of FlexLLM, which clearly stated its goal: to dynamically reallocate resources for inference workloads under high-traffic conditions, given the strict latency requirements. In contrast, this paper does not articulate why unifying fine-tuning and inference is important, what problem it aims to solve, or how success should be measured.

The paper claims 3× performance improvement and 46.4× SLO attainment, but fails to explain whether these improvements are addressing a critical bottleneck. Is the 3× performance gain addressing a real, high-priority need? Is the 46.4× SLO improvement sufficient to meet practical latency constraints? Without clearly stated goals, the work risks appearing like it "shot first and drew the target afterward."

**Dangling table:**

Table 1 is presented without any explanation of its purpose or guidance on how to interpret it.

**Insufficient experimental details:**

For example, as noted in the FlexLLM paper, ShareGPT data lacks critical metadata such as request arrival times. FlexLLM addressed this issue with corrective measures. This paper does not mention whether this limitation was considered, or whether any mitigation strategies were applied.

### Lack of Motivation

(See above.) Without a well-articulated motivation, it’s difficult to assess the importance or relevance of the proposed solution.

### Poor Experimental Design

The paper avoids any meaningful comparison with FlexLLM, claiming that it "is trapped in an infinite loop while loading LoRA models under multi-LoRA inference." However:

FlexLLM was not designed for multi-LoRA, so it's not fair to disqualify it based on that alone.

The absence of a direct comparison should not prevent using proxy evaluations, especially for latency and resource efficiency.

FlexLLM may have lower resource efficiency but better latency, which aligns with its design objective. The paper fails to explore or even hypothesize on such trade-offs.

### Poor Baseline Selection

Since FlexLLM is deemed "infeasible," the only baseline considered is PEFT, which is not designed for co-serving fine-tuning and inference tasks. This makes PEFT an overly simplistic and weak baseline.

The related work section cites several relevant approaches, but none of them appear to have been included as competitive baselines in the experiments.

### Poor Workload Design

As mentioned earlier, the ShareGPT dataset lacks critical information such as request arrival times, making it difficult to evaluate the realism or representativeness of the simulated workload. Without addressing this, the trustworthiness of the experimental results is significantly weakened.

---

> ### Author Rebuttal · Authors · 2025-07-31
>
> Thank you for your detailed review of our work. Before responding to the weaknesses and questions you raised. We clarify our research background, scenario definition, and metrics used to measure results.
>
> # Background, Scenario Definition, and Key Experimental Metrics
>
> Loquetier is a framework for unified fine-tuning and serving of multiple LoRAs. As mentioned in the Introduction, the research background is that LoRA and its related efficient fine-tuning techniques have demonstrated excellent performance across various specialized tasks and show significant potential in many tasks, such as personalization and user preference-related tasks, leading to their increasingly widespread adoption and promising future applications. However, the hardware resources available for LLM training, fine-tuning, and serving remain extremely limited. This leads to our application scenario: how to improve the efficiency of multi-LoRA fine-tuning and inference tasks under constrained hardware conditions.
>
> Specifically, most existing system frameworks require a hard switch when handling different tasks, meaning they must first stop the current task before starting another. Such switches consume significant time and require saving fine-tuning states and migrating inference requests, which occupy substantial bandwidth and storage space. During downtime, the occupied computational resources cannot be utilized, leading to a significant decline in the system's hardware resource utilization efficiency.
>
> Unifying fine-tuning and inference of multiple LoRAs upon one base model can address the aforementioned issues and improve the efficiency of multi-LoRA fine-tuning and inference tasks. Integrating the two distinct tasks is a practical approach, as the only difference between them lies in the gradient calculation and parameter updates during the backpropagation process. Regardless of the task type, sharing the same base model can significantly reduce memory requirements.
>
> According to the description in the Experiments section, we measure the efficiency of these two tasks as follows, and list the meanings of all experimental metrics in Appendix B:
>
> The efficiency metric for the fine-tuning task is the throughput of fine-tuning and evaluation, i.e., the number of tokens fine-tuned and evaluated per unit of time, which is equivalent to the time required to fine-tune with the same data and configuration.
>
> The efficiency metric for the inference task is inference throughput, which is the number of tokens inferred per unit of time. Additionally, serving metrics targets are used to define the availability of the service system, so the secondary metric is the service level objective attainment (rate). Appendix C clarifies the service metric settings, including the maximum request wait time, maximum decoding latency per token, and average decoding latency. By calculating whether each request meets these strict serving metrics, our experimental results have sufficient practical significance.
>
> # Motivation and Objectives
>
> Our motivation and objectives are mentioned in the paper and the first section "Background, Scenario Definition, and Key Experimental Metrics" in response. Our work is not an extension of FlexLLM but rather an optimization of unified fine-tuning and inference work in the rapidly growing and highly promising LoRA domain. We also present experimental metrics for evaluating these tasks and compare our results with other works under the same conditions, demonstrating significant improvements achieved by our approach.
>
> Regarding Table 1 in the paper, we naturally use symbols and footnotes to indicate whether the baseline can complete each scenario. We use $\checkmark$ to indicate that it can run in the corresponding scenario, $\triangle$ means it encountered some issues that caused the test to fail but can be achieved by degrading to a more basic method, and $\times$ refers that it cannot run in that scenario.
>
> Regarding the high-priority requirements you mentioned, according to technical reports, official blogs, and other customer cases from leading internet companies and cloud platforms, such differentiated services are typically provided at the application layer, including real-world scenarios involving the scalable use of LoRA. Our work is a unified underlying framework for fine-tuning and inference tasks, designed to efficiently handle incoming requests to the best of its ability. Loquetier is compatible with various cluster management frameworks and business frameworks, so such requirements can be fulfilled through collaboration with these systems. Based on related work involving multi-LoRA or multi-model parallel inference, including FlexLLM which adopts Orca's research findings, no priority differentiation or other differentiated services have been implemented. As mentioned earlier, this is not the responsibility of the underlying framework.
>
> # Experimental Details Related to FlexLLM
>
> FlexLLM is the first work in the field of unified fine-tuning and inference for multi-PEFT models. We fully acknowledge its pioneering nature and have never denied its contributions in our paper. FlexLLM still has limitations in the specific scenarios it claims to support, which is precisely why we should use it as a critical baseline for comparison. We were also surprised by FlexLLM's performance in multi-LoRA inference scenarios, but our experimental results indicate that it cannot support multi-LoRA inference in a single runtime instance.
>
> Please allow us to update the performance of FlexLLM in multiple LoRA inference scenarios. We conducted a smaller-scale test, and the results showed that FlexLLM ran for several hours while repeatedly loading LoRA in a loop for a test designed for 5 minutes. We have not studied its underlying implementation in detail, so we will not speculate on the cause of this phenomenon.
>
> We will conduct additional tests to fully demonstrate FlexLLM's performance in multi-LoRA inference scenarios. We will provide feedback on the experimental results as soon as these tests are completed.
>
> Regarding the trade-off between resource efficiency and latency, we will elaborate further from the following two aspects:
>
> As mentioned in the first section, we are more concerned with the performance of multi-LoRA tasks when a large number of requests arrive under limited hardware resources. As stated in your review comments, this differs from FlexLLM's objectives. We have outlined the experimental metrics in Appendix B of the paper and in the first section in our response, and these metrics comprehensively reflect whether we have achieved our stated objectives.
>
> Additionally, we mentioned the issue of differentiated services in the “Motivation and Objectives” section. Since altering the request arrival rate of inference tasks also affects latency, we believe this conclusion also applies to differentiated services.
>
> We have still managed to conduct inference latency evaluations in both single-LoRA and multi-LoRA scenarios to address your concerns. However, our background and scenario definitions are focused on the fine-tuning and inference of multiple LoRA models, so the comparison of single-LoRA inference scenarios is not our primary focus.
>
> In a single LoRA scenario, FlexLLM achieves 60 tokens/s (tps), corresponding to a latency of 17ms, while Loquetier reaches 30tps and 33ms. In a multi-LoRA scenario (4 LoRA models with 1 infer request per LoRA), FlexLLM achieves 38 tps and 105ms, while Loquetier reaches 120 tps and 33ms. The Peft baseline lags significantly behind in both scenarios. To better address the scenarios we are discussing, Loquetier's design approach remains the optimal solution at present.
>
> # Baselines
>
> We have received valuable suggestions from the reviewers regarding the evaluation. We would like to provide additional explanations and the derived results to address the issues raised.
>
> For S-LoRA, FlexLLM have compared with it. S-LoRA only optimized for inference tasks and performs worse than FlexLLM in fine-tuning and inference tasks; FlexLLM's experiments include the Peft baseline for fine-tuning tasks, and our results in similar experimental settings are better than the equivalent fine-tuning performance of Peft in the S-LoRA+Peft baseline in FlexLLM's experiments, while the performance of the Peft is better than that of FlexLLM in several experimental environments (FlexLLM has not tested a direct comparison with Peft when only performing fine-tuning tasks; based on the result in their paper and the number of GPUs used, we can infer that Peft's fine-tuning efficiency is higher than that of FlexLLM under the same hardware resources). Therefore, based on the proxy evaluation method, we can also confirm that our performance is better than that of FlexLLM and S-LoRA. Additionally, S-LoRA significantly outperforms vLLM in its multi-LoRA inference experiments, and its experimental environment is comparable to ours. Furthermore, S-LoRA does not support newer models, including Llama 3. For these reasons, we did not design experiments comparing S-LoRA and vLLM.
>
> As mentioned in the Introduction, there are very few training-inference integrated frameworks that support multi-LoRA. FlexLLM may be the only proper baseline. The other methods mentioned in the Related work are mostly general optimizations for base models or further improvements to the LoRA method itself, and they are not suitable for evaluation as test baselines; our work is also compatible with these methods.
>
> # Workload Design
>
> As for the metadata you mentioned that FlexLLM created from third-party sources, this was added in the version of the paper submitted in May 2025. When we were working on Loquetier, we referred to the only version available at the time (the first version in Feb 2024). We'd rather compare it to that version, but we are open to further discussion on the points you raised. Please let us know if you would like to discuss this part further.

---

> > ### Comment · Reviewer_WdRD · 2025-08-03
> >
> > > In a single LoRA scenario, FlexLLM achieves 60 tokens/s (tps), corresponding to a latency of 17ms, while Loquetier reaches 30tps and 33ms. In a multi-LoRA scenario (4 LoRA models with 1 infer request per LoRA), FlexLLM achieves 38 tps and 105ms, while Loquetier reaches 120 tps and 33ms. The Peft baseline lags significantly behind in both scenarios. To better address the scenarios we are discussing, Loquetier's design approach remains the optimal solution at present.
> >
> > Thank you for providing the data points. Could you please elaborate on the resources involved in each case, particularly in terms of GPUs and memory usage?
> >
> > > Therefore, based on the proxy evaluation method, we can also confirm that our performance is better than that of FlexLLM and S-LoRA.
> >
> > From my understanding, this analysis focuses only on fine-tuning tasks. Could you also clarify the implications for inference and mixed workloads? It would be highly valuable to include summary tables comparing these scenarios in the next round of communication.
> >
> > > When we were working on Loquetier, we referred to the only version available at the time (the first version in Feb 2024). We'd rather compare it to that version, but we are open to further discussion on the points you raised. Please let us know if you would like to discuss this part further.
> >
> > Lastly, I believe that request arrival times are a critical factor. I would appreciate it if the authors could provide empirical results addressing this aspect as well.

---

> > > ### Author Response · Authors · 2025-08-04
> > >
> > > > Could you please elaborate on the resources involved in each case, particularly in terms of GPUs and memory usage?
> > >
> > > We used H800 for testing, providing the full 80G of GPU memory available for each test. Loquetier used 16.10G / 16.21G of GPU memory in single LoRA and multi-LoRA inference, respectively. FlexLLM did not provide a way to directly view GPU memory usage. It would occupy slightly more GPU memory than the set value, regardless of whether it was actually needed. We gradually lowered the value, and the minimum GPU memory FlexLLM could run on in this test was 17.54G / 17.93G. This may be because FlexLLM needs to reserve space for some kernel computations, but the space it requires is greater than that of Loquetier. Additionally, in terms of GPU utilization, the SM utilization of them is also similar (around 66%).
> > >
> > > > Could you also clarify the implications for inference and mixed workloads? It would be highly valuable to include summary tables comparing these scenarios in the next round of communication.
> > >
> > > Of course, we would like to clarify this point further.
> > >
> > > FlexLLM's inference performance in the scenarios has already been explained in our previous response. Based on the proxy evaluation of fine-tuning tasks in our response, it can be inferred that in multi-LoRA scenarios, Loquetier also outperforms FlexLLM in unified fine-tuning and inference tasks.
> > >
> > > According to FlexLLM's tests (specifically, their paper in the first version), the experiments compared the performance of S-LoRA+Peft's baseline with FlexLLM in fine-tuning and inference tasks, and also showed the number of GPUs required by S-LoRA at different request arrival rates in its charts. As the arrival request rate increases, Peft's fine-tuning speed decreases significantly, which means that S-LoRA uses more computing resources. In comparisons of each setting, S-LoRA and FlexLLM should have similar inference performance to meet the requirements of the inference task, and this performance covers scenarios involving inference-only and mixed workloads. Since Loquetier performs better than FlexLLM in these scenarios, Loquetier should also outperform S-LoRA in scenarios involving inference-only and unified fine-tuning and inference, as well as outperform vLLM, which showed poorer results in experiments described in the S-LoRA paper.
> > >
> > > We are currently running independent experiments on S-LoRA in inference task scenarios. We will provide feedback on the results as soon as the experiments are completed.
> > >
> > > > Lastly, I believe that request arrival times are a critical factor. I would appreciate it if the authors could provide empirical results addressing this aspect as well.
> > >
> > > Considering that this is a more complex topic, we will provide a more detailed response in our next comment.

---

> > > ### Author Response · Authors · 2025-08-04
> > >
> > > > Lastly, I believe that request arrival times are a critical factor. I would appreciate it if the authors could provide empirical results addressing this aspect as well.
> > >
> > > Regarding request arrival times, the objective is to verify whether the framework can adapt to different request arrival rates and meet predetermined goals in dynamic input scenarios.
> > >
> > > We designed a unified fine-tuning and inference task within hardware constraints for dynamic input scenarios (Section 4.2 - Unified Fine-tuning and Inference). Experimental results show that Loquetier can handle any dynamic input changes under hardware limitations.
> > >
> > > Additionally, request arrival times and other metadata, such as which user and session the request originated from, pose challenges in how to distribute requests across multiple devices and validate the effectiveness of scheduling strategies in multi-machine and cross-device scenarios.
> > >
> > > We would like to point out first that, based on the relevant research we have collected, the inherent nature of the LoRA method precludes these parallel strategies in most cases, and there has been little targeted optimization for strategies such as tensor parallelism when using LoRA, especially in fine-tuning tasks, as the LoRA computation itself already has no further room for parallel optimization. The LoRA method reduces fine-tuning and model storage overhead by decreasing the number of parameters in fine-tuning, but the computations introduced by its forward and backward processes increase communication overhead during fine-tuning and inference in the aforementioned cross-GPU scenarios. For example, for each LoRA linear layer, the communication overhead required for synchronization during fine-tuning reaches $2 \times s \times (r + h) \times n$, where $n$ is the number of GPUs involved in parallelism. Considering the number of LoRA layers and the dimension of hidden_size in large models, the communication overhead is extremely high compared to the benefits. Therefore, we expect the model to be loaded into a single GPU. For strategies like pipeline parallelism, data only needs to be transferred once between two layers on different devices, so our work fully supports this parallel strategy.
> > >
> > > Loquetier's computation process can be split using all-reduce / all-gather. This means that, if necessary, we can also run larger models in a multi-GPU environment without incurring any additional overhead compared to existing methods.
> > >
> > > As described above, our current goal is to achieve a professional single-machine framework that unifies multi-LoRA fine-tuning and inference scenarios. Loquetier is designed for single-device while retaining sufficient flexibility for scheduling and other strategies. For multi-device scenarios, as described in Section 1 “Motivation and Objectives” in our first response and the related literature review, this is handled by the application-level business framework and cluster management system. Single-machine system frameworks should generally not interfere with this part of scheduling, as this would hinder the flexibility of application-level frameworks, reduce scheduling space, and limit the range of available scheduling strategies. Loquetier's relevant data remains fully transparent, enabling collaboration with various cluster management frameworks and business frameworks through information such as the number of requests on each device, task progress for fine-tuning and inference, and loaded available LoRAs. As we just mentioned, Loquetier is compatible with these application-level frameworks and can collaboratively provide efficient cluster-level services.
> > >
> > > For example, the cluster system can distribute incoming new requests based on the LoRA and load running on each Loquetier instance, store the most recent KV cache in device memory, and attempt to distribute requests from the same user to the previously assigned device. When migration is required, multi-process methods can be used to simultaneously perform unified inference and fine-tuning tasks, and request migration, and can be combined with other migration optimization tasks to reduce migration latency.
> > >
> > > For single-machine system frameworks that do not expose this information and multi-machine frameworks, they typically adopt a closed-system design, where external systems cannot access relevant critical data, necessitating the implementation of some scheduling frameworks. This design results in a multi-level scheduling system that becomes increasingly complex, with different layers not open to each other, leading to less efficient scheduling strategies. In large-scale applications, this is less effective than a transparent multi-level scheduling system where different layers are open to each other and a single-machine system that collaborates with application-layer scheduling frameworks.

---

> > > > ### Comment · Reviewer_WdRD · 2025-08-05
> > > >
> > > > Thank you for the comprehensive clarification. However, I’m afraid my concerns remain unaddressed.
> > > >
> > > > The paper claims that:
> > > >
> > > > > FlexLLM is the first work in the field of unified fine-tuning and inference for multi-PEFT models.
> > > >
> > > > Given this, I believe greater emphasis should be placed on the **empirical results** supporting unified fine-tuning and inference. Yet, the main empirical point provided is:
> > > >
> > > > > it can be inferred that in multi-LoRA scenarios, Loquetier also outperforms FlexLLM in unified fine-tuning.
> > > >
> > > > This indirect inference is insufficient to substantiate the claim.
> > > >
> > > > Additionally, when I first read **Unified Fine-tuning and Inference** in Section 4.2, my primary concern was the **realism and representativeness of the simulated workload**. Unfortunately, no new evidence has been presented to address this issue.

---

> > > > > ### Author Response · Authors · 2025-08-07
> > > > >
> > > > > We appreciate and respect your review. Thank you for sharing your concerns, opinions, and suggestions with us based on your professional knowledge.
> > > > >
> > > > > Based on the reviewers' suggestions, we have completed the main part of the FlexLLM experiment, and we would like to share the results with you.
> > > > >
> > > > > In our paper, we mentioned that FlexLLM cannot run fine-tuning tasks in their official Docker environment, as it is missing at least one operator in the backward computation process. After investigation, we confirmed that the missing operator is OP_SCALAR_TRUE_DIV. Additionally, we noted that five other operators from the same source (the header file element_unary.h) are also missing: OP_GELU, OP_RELU, OP_SIGMOID, OP_TANH, and OP_ELU.
> > > > >
> > > > > Before submitting the paper, we spent a considerable amount of time attempting to build their framework, but our efforts ended in failure. Therefore, we only included results of FlexLLM in certain scenarios in the paper. Over the past few days, we attempted to build dozens of commits dating back to Feb 2024 (FlexLLM paper v1 published). encountering compilation and runtime issues such as source code errors, dependency errors, segmentation faults, and PEFT model incompatibility. In all these versions, instances of the aforementioned operators were never included in the backward computation process.
> > > > >
> > > > > We noticed that the FlexLLM repository contained forward and backward kernels related to element_unary, but they had never applied these backward kernels to fused.cu, resulting in its inability to perform fine-tuning tasks. **We ultimately built a runnable version by instantiating these kernels at the missing locations. This fix was based on our understanding of the FlexLLM framework, and we did not implement any additional computational steps. The correction will not result in a performance degradation of FlexLLM.** As mentioned earlier, we want you to know that we had previously made every effort to reproduce its functionality, and the reason for the failure to reproduce was not due to us.
> > > > >
> > > > > The experiment comparison results for fine-tuning and unified tasks including FlexLLM are as follows; all experimental settings (including metric settings) are consistent with those in the paper; other data are taken from the experimental results in the paper:
> > > > >
> > > > > ## Finetuning
> > > > >
> > > > > | | FlexLLM | | | | Loquetier | | | | PEFT | | | |
> > > > > | - | - | - | - | - | - | - | - | - | - | - | - | - |
> > > > > | Setup | Total Time (s) | FTime (s) | ETime (s) | FTP (tps) | Total Time | FTime | ETime | FTP | Total Time | FTime | ETime | FTP |
> > > > > | S | 2261.28 | 2132.35 | 128.93 | 3841.77 | 1955.14 | 1805.75 | 149.39 | 4536.62 | 2481.63 | 1726.17 | 755.46 | 4745.76 |
> > > > > | M | 2262.46 | 2135.99 | 126.47 | 3835.22 | 1851.45 | 1765.75 | 85.7 | 4639.4 | 2480.78 | 1724.94 | 755.84 | 4749.16 |
> > > > >
> > > > > ## Unified
> > > > >
> > > > > | | | FlexLLM | | | Loquetier | | | PEFT | | |
> > > > > | - | - | - | - | - | - | - | - | - | - | - |
> > > > > | Setup | RPS | FTP (tps) | DTP (tps) | SLO (%) |  FTP | DTP | SLO | FTP | DTP | SLO |
> > > > > | S | 1 | 990.58 | 66.13 | 8.5 | 2246.14 | 268.9 | 100 | 3916.88 | 9.09 | 7.06 |
> > > > > | S | 3 | 988.42 | 55.32 | 1.11 | 1724.66 | 610.54 | 92.06 | 3802.13 | 21.96 | 5.29 |
> > > > > | S | 5 | 994 | 69.88 | 0.37 | 1487.4 | 600.87 | 62.67 | 3727.26 | 26.95 | 4.01 |
> > > > > | M | 1 | 958.38 | 25 | 6.17 | 2256.49 | 283.37 | 100 | 4039.2 | 3.18 | 2.34 |
> > > > > | M | 3 | 1059.39 | 27.63 | 1.11 | 1739.41 | 619.66 | 92 | 3974.24 | 8.12 | 2.11 |
> > > > > | M | 5 | 1063.36 | 134.72 | 0.57 | 1505.21 | 601.2 | 56.07 | 3920.17 | 7.52 | 1.6 |
> > > > >
> > > > > * All experiments used Partial (which means 3 target modules).
> > > > > * In Setup, S/M denotes single/multiple LoRA; S/M in unified tasks denotes the number of LoRA in inference tasks, while fine-tuning tasks are all single LoRA.
> > > > > * tps denotes tokens per second, FTime/ETime denotes Fine-tuning/Evaluation Time, FTP/DTP denotes Fine-tuning/Decoding throughput, SLO denotes Service Level Objective (Attainment).
> > > > > * Configurations that appear in the paper but are not listed in the table above are not supported by FlexLLM. The issues with target modules have been identified in the paper, and the relevant experiments in the unified task can only perform single LoRA fine-tuning because FlexLLM cannot perform multi-LoRA fine-tuning.
> > > > > * FlexLLM will only execute one fine-tuning task at a time, even if two fine-tuning requests are submitted simultaneously; the fine-tuning instances of FlexLLM consume far more than 40G of GPU memory (H800 has 80G). This means that in our scenarios, similar to PEFT, FlexLLM can only run multiple fine-tuning requests sequentially. The data in the table is the result of sequential runs.
> > > > >
> > > > > The empirical results above are consistent with our previous findings, namely that Loquetier outperforms FlexLLM in these scenarios.
> > > > >
> > > > > Based on previous proxy evaluations and the results above, we are even more confident that Loquetier outperforms the S-LoRA+PEFT baseline. We will complete the remaining experiments as soon as possible and share the results.

---

> ### Comment · Reviewer_WdRD · 2025-08-07
>
> Thank you to the authors for the great effort during the rebuttal phase. It's encouraging to see empirical results supporting **Unified Fine-tuning and Inference**.
>
> > and the reason for the failure to reproduce was not due to us
>
> To clarify, it was never my intention to blame the authors for the failures of previous approaches, nor did I suggest that the authors were responsible for fixing those failures. My original suggestion was to explore reasonable _proxy evaluations_. In retrospect, there may have been a misunderstanding—by proxy evaluations, I did not mean qualitative assessments or indirect inferences, but rather empirical results that could closely reflect FlexLLM’s performance in cases where direct reproduction was not possible. That said, it's great to see the authors were able to work through this issue.
>
> The current empirical results are solid, and I am raising my score to 3. The last missing piece, in my view, is the **BurstGPT** data. While the authors designed their workload to follow known patterns, subtle differences may exist that are not captured by available statistical data. Additionally, the segmentation into morning, afternoon, and overnight periods feels overly coarse—I’m not yet convinced it adequately reflects fine-grained workload fluctuations.
>
> I've tried to respond as quickly as possible to leave the authors enough time to incorporate these points before the rebuttal phase concludes.

---

> ### Author Response · Authors · 2025-08-08
>
> > I've tried to respond as quickly as possible to leave the authors enough time to incorporate these points before the rebuttal phase concludes.
>
> Thank you for giving us as much time as possible to provide further responses.
>
> Based on your suggestion, we conducted another dynamic workload experiment using the BurstGPT dataset. We are willing to include this experiment and the previous experimental results on FlexLLM in the paper. The specific data and results are as follows:
>
> The BurstGPT dataset includes 60 days of request arrival times, input lengths, and output lengths. According to the description in the BurstGPT paper, their testing tool provides sampling slices with a duration of 20 minutes. The latest version of FlexLLM paper used a single 20-minute sampling duration for testing.
>
> **We aim to demonstrate Loquetier's adaptability to dynamic workload as comprehensively as possible. Therefore, we sampled a total of 6 time periods, each lasting 20 minutes, resulting in a total simulated duration of 2 hours.** Specifically:
>
> - 1 slot is a low-load time slot (average RPS < 1);
> - 2 slots are medium-load time slots (average RPS between 1 and 2);
> - 3 slots are high-load time slots (average RPS > 2).
>
> Based on our statistics of the BurstGPT dataset, less than one-third of the time slots are high-load time slots, with the majority being low-load time slots. Since low-load time slots may be considered less challenging for the framework, we adopted the above settings.
>
> The number of requests for the 6 time slots are as follows: 676, 2145, 1465, 2823, 2360, and 1856.
>
> Since NeurIPS does not allow the upload of images or links, we regret that we are unable to show you the charts, as this is the most intuitive way to present the experimental results. We hope to describe the chart content as clearly as possible in writing. Our statistical table is as follows:
>
> | Time Period (min) | Inference Throughput (tps) | Finetuning Throughput (tps) |
> | - | - | - |
> | 0-20 | 120 (80-300) | 2500(2300-3000) |
> | 20-25 | 580 (550-610) | 1600 (1500-2000) |
> | 25-40 | 135 (80-330) | 2400 (2300-2800) |
> | 40-60 | 280 (200-430) | 2000 (1650-2400) |
> | 60-80 | 600 (570-630) | 1500 (1450-1600) |
> | 80-100 | 400 (250-600) | 1850 (1500-2150) |
> | 100-120 | 280 (120-540) | 2150 (1600-2700) |
>
> * tps denotes token per second.
>
> In the first 20 minutes, the low load enabled the fine-tuning task to achieve a throughput of 2,500 tps, which is consistent with the performance observed in the initial phase of the dynamic workload experiment described in our paper.
>
> Between 20 and 25 minutes, there was a sudden spike in high load, with the highest peak RPS reaching 11. At this point, the inference throughput rapidly increased to approximately 580 tps, while the fine-tuning task yielded to it, dropping to approximately 1,600 tps. After the 25th minute, the rate of incoming inference requests decreased, and the fine-tuning throughput returned to its previous level.
>
> Starting at the 40-minute mark, moderate load caused the inference task's throughput to fluctuate around 280±100 tps, while fine-tuning throughput decreased to around 2000 tps.
>
> Starting at the 60th minute, continuous high load for 40 minutes caused the inference throughput to exceed 600 tps at one point. Subsequently, the request arrival rate fluctuated significantly, and Loquetier's inference and fine-tuning throughput fluctuated accordingly (with an amplitude of approximately ±50 tps), indicating that our work can respond promptly to high-frequency load changes.
>
> In the final 20 minutes of the medium-load scenario, the request arrival rate still fluctuated significantly, causing the throughput of both tasks to continue oscillating. The inference throughput gradually decreased as the load decreased, while the fine-tuning throughput gradually increased as computational resources were released.
>
> The final result of SLO Attainment is 92.37%, with all requests that failed to meet service metrics occurring during workload spikes under high workload (RPS>5). Based on previous experiments and responses, this exceeded the hardware load limit. During other periods, Loquetier was able to meet our objectives.

---

> > ### Comment · Reviewer_WdRD · 2025-08-09
> >
> > Thank you—the results are solid, and all of my concerns have been addressed. I will raise my score accordingly.

---

> > > ### Author Response · Authors · 2025-08-09
> > >
> > > Thank you very much for your valuable comments and suggestions! We will incorporate all the additional experiments in our next revision to make the paper more convincing for the readers.

---

### Official Review · Reviewer_fRyY · 2025-07-01

**Clarity:** 3
**Significance:** 2
**Originality:** 2
**Rating:** 4
**Confidence:** 3

**Summary:**

The authors propose Loquetier, a framework for serving and fine-tuning multiple LoRA models simultaneously. The authors describe the workings of their framework and compare its performance with other setups for training and evaluating LoRA fine-tuned models.

**Questions:**

- I would be grateful if the authors could address my comments regarding comparisons with other methods (adding more baselines in each of the serving and fine-tuning settings).

- I would also like to know if they were able to bypass the issues presented during their previous iterations with FlexLLM, by building the tool from scratch.

Improving upon both of these points would be great for the paper, and I would consider increasing my score.

**Ethical Concerns:**

["NO or VERY MINOR ethics concerns only"]

**Final Justification:**

Despite my initial concerns about this work, the authors have put a lot of effort into addressing both mine and other Reviewers' concerns, especially regarding comparisons with FlexLLM. The additional experimental results support the usefulness of this framework, and thus I think that most of my concerns have been addressed. While there are still only a few comparisons with existing baselines, this can be attributed to the fact that the framework proposed by the authors tackles a unique problem in Multi-LoRA model serving.

**Limitations:**

The authors have adequately addressed the limitations and impact of their work.

**Paper Formatting Concerns:**

No major concerns.

**Quality:**

2

**Strengths And Weaknesses:**

Strengths

- The paper is clearly written and easy to follow. I didn’t have any issues with following the ideas presented or the arguments made by the authors.

- The proposed framework is also quite comprehensive. Being able to accommodate a variety of use cases is an important ability for a good framework. The fact that Loquetier supports both single and multi LoRA serving and fine-tuning makes it suitable for use in a variety of contexts.

- The structure of the framework itself is solid. The use of the virtualized module as described by the authors serves well to reduce the memory footprint of the model and allow for easier sharing of components across different LoRA instances.

Weaknesses

- My main concern with this paper is the benchmarking of Loquetier. Currently the only meaningful comparison made is to PEFT (as results with FlexLLM are really only presented in the single LoRA serving case, as seen in Table 2). Given that PEFT is very simple to begin with, I am not sure how well Loquetier compares to other frameworks. For the serving case at the very least, the authors should compare to more frameworks, such as VLLM and S-LoRA, to make comparisons more convincing (and ideally more frameworks should be added in the fine-tuning case as well).

- Regarding FlexLLM, I am a bit surprised that results are not presented for the single LoRA fine-tuning case, as this is a mode that is explicitly presented. The authors mention that an unsupported operation prevented them from evaluating this case. In Section 4.1, the authors mention that they used prebuilt docker images to use FlexLLM. I think that the unsupported operation should be avoided if FlexLLM is built from scratch instead. If building the framework from scratch has not been done, I think it should be since that should allow for FlexLLM to be properly included as a comparison framework.

- On a more minor note, I understand the main technical novelty to be the virtualized module. If so, the authors should highlight this a bit more and also directly contrast choices made by other frameworks in Section 3. Doing so would allow for the technical novelty of the paper to be better understood.

**Post Rebuttal Comments**: As noted below, the additional experimental results included in the rebuttal strengthen the paper, and I have thus raised my score.

---

> ### Author Rebuttal · Authors · 2025-07-31
>
> Thank you for your detailed review of our work. Our responses to the weaknesses and questions you raised are as follows:
>
> # Baselines
>
> We have received valuable suggestions from the reviewers regarding the evaluation. We would like to provide additional explanations and the derived results to address the issues raised.
>
> For S-LoRA, FlexLLM have compared with it. S-LoRA is only optimized for inference tasks and performs worse than FlexLLM in fine-tuning and inference tasks; FlexLLM's experiments include the Peft baseline for fine-tuning tasks, and our results in similar experimental settings are better than the equivalent fine-tuning performance of Peft in the S-LoRA+Peft baseline in FlexLLM's experiments, while the performance of the Peft is better than that of FlexLLM in several experimental environments (FlexLLM has not tested a direct comparison with Peft when only performing fine-tuning tasks; based on the result in their paper and the number of GPUs used, we can infer that Peft's fine-tuning efficiency is higher than that of FlexLLM under the same hardware resources). Therefore, based on the proxy evaluation method, we can also confirm that our performance is better than that of FlexLLM and S-LoRA. Additionally, S-LoRA significantly outperforms vLLM in its multi-LoRA inference experiments, and its experimental environment is comparable to ours. Furthermore, S-LoRA does not support newer models, including Llama 3. For these reasons, we did not design experiments comparing S-LoRA and vLLM.
>
> As mentioned in the Introduction, there are very few training-inference integrated frameworks that support multi-LoRA. FlexLLM may be the only proper baseline. The other methods mentioned in the Related work are mostly general optimizations for base models or further improvements to the LoRA method itself, and they are not suitable for evaluation as test baselines; our work is also compatible with these methods.
>
> # Further Attempts on FlexLLM testing
>
> For the performance of FlexLLM in multiple LoRA inference scenarios. We conducted a smaller-scale test, and the results showed that FlexLLM ran for several hours while repeatedly loading LoRA in a loop for a test that should not have lasted more than 5 minutes—it indeed did not get stuck in an infinite loop. We have not studied its underlying implementation in detail, so we will not speculate on the cause of this phenomenon.
>
> We will further attempt to rebuild FlexLLM from scratch based on more commits, which will take some time. We will provide feedback on the results after thorough testing. However, as mentioned in the paper, FlexLLM is missing a critical operation operator (the case for OP_CONV2D in FusedOp::peft_bwd_task), and we have not found any implementation records for this operator in historical commits. Therefore, we do not expect the built version to support LoRA fine-tuning functionality.
>
> # Comparison of Virtualized Module and Selections from Other Works
>
> The implementation of PEFT directly modifies the base model, thereby rendering it non-reusable once processed. PEFT adopts a high-coupling design for LoRA, incorporating multiple parameters from different LoRAs within the same linear layer module. For different efficient fine-tuning methods, it designs the MixedModel module and related modules, enumerating and storing each supported efficient fine-tuning method for storage and invocation, causing different methods mixed together, resulting in extremely high coupling and poor maintainability.
>
> For other frameworks, we categorize them into two types. The first type involves frameworks designed from scratch and built from the ground up. Since these frameworks do not need to consider compatibility issues, they generally do not retain the original structure of the base model, as the framework can handle structural modifications. The second category includes systems like S-LoRA and Punica, which merge multiple LoRA models for processing and directly modify the original model structure. This not only renders the original model unusable by other methods but also couples different LoRA models together, thereby reducing operability.

---

> > ### Comment · Reviewer_fRyY · 2025-08-04
> > **Re: Rebuttal**
> >
> > Thank you very much for your response to mine and the other Reviewers' concerns. I would be grateful if you could update us further on performance of FlexLLM in the multi-LoRA setting, as well as whether you were able to compare in the single LoRA finetuning setting as well (although I am puzzled by the unsupported operation encountered - I might be missing something, but I don't understand where 2-dimensional convolution is needed for single-LoRA finetuning for the Llama-3 architecture, which seems to be the missing operation implied by the error).
> >
> > I am inclined to raise my score slightly, given that this is a unique setting without many different baselines. At the same time, the small amount of comparisons makes it difficult to accurately judge the novelty of Loquetier. I think if the comparisons with FlexLLM were cleaner (instead of it appearing impossible to compare against it in most settings) then the paper would be much stronger.

---

> > > ### Author Response · Authors · 2025-08-07
> > >
> > > Thank you for your patience. We have completed the main part of the FlexLLM experiment and would like to share the results with you.
> > >
> > > In our paper, we mentioned that FlexLLM cannot run fine-tuning tasks in their official Docker environment, as it is missing at least one operator in the backward computation process. After investigation, we confirmed that the missing operator is OP_SCALAR_TRUE_DIV (the previously mentioned OP_CONV2D belongs to another operator enumeration type). Additionally, we noted that five other operators from the same source (the header file element_unary.h) are also missing: OP_GELU, OP_RELU, OP_SIGMOID, OP_TANH, and OP_ELU.
> > >
> > > Before submitting the paper, we spent a considerable amount of time attempting to build their framework, but our efforts ended in failure. Therefore, we only included results of FlexLLM in certain scenarios in the paper. Over the past few days, we attempted to build dozens of commits dating back to Feb 2024 (FlexLLM paper v1 published). encountering compilation and runtime issues such as source code errors, dependency errors, segmentation faults, and PEFT model incompatibility. In all these versions, instances of the aforementioned operators were never included in the backward computation process.
> > >
> > > We noticed that the FlexLLM repository contained forward and backward kernels related to element_unary, but they had never applied these backward kernels to fused.cu, resulting in its inability to perform fine-tuning tasks. **We ultimately built a runnable version by instantiating these kernels at the missing locations. This fix was based on our understanding of the FlexLLM framework, and we did not implement any additional computational steps. The correction will not result in a performance degradation of FlexLLM.** As mentioned earlier, we want you to know that we had previously made every effort to reproduce its functionality, and the reason for the failure to reproduce was not due to us.
> > >
> > > The experiment comparison results for fine-tuning and unified tasks including FlexLLM are as follows; all experimental settings (including metric settings) are consistent with those in the paper; other data are taken from the experimental results in the paper:
> > >
> > > ## Finetuning
> > >
> > > | | FlexLLM | | | | Loquetier | | | | PEFT | | | |
> > > | - | - | - | - | - | - | - | - | - | - | - | - | - |
> > > | Setup | Total Time (s) | FTime (s) | ETime (s) | FTP (tps) | Total Time | FTime | ETime | FTP | Total Time | FTime | ETime | FTP |
> > > | S | 2261.28 | 2132.35 | 128.93 | 3841.77 | 1955.14 | 1805.75 | 149.39 | 4536.62 | 2481.63 | 1726.17 | 755.46 | 4745.76 |
> > > | M | 2262.46 | 2135.99 | 126.47 | 3835.22 | 1851.45 | 1765.75 | 85.7 | 4639.4 | 2480.78 | 1724.94 | 755.84 | 4749.16 |
> > >
> > > ## Unified
> > >
> > > | | | FlexLLM | | | Loquetier | | | PEFT | | |
> > > | - | - | - | - | - | - | - | - | - | - | - |
> > > | Setup | RPS | FTP (tps) | DTP (tps) | SLO (%) |  FTP | DTP | SLO | FTP | DTP | SLO |
> > > | S | 1 | 990.58 | 66.13 | 8.5 | 2246.14 | 268.9 | 100 | 3916.88 | 9.09 | 7.06 |
> > > | S | 3 | 988.42 | 55.32 | 1.11 | 1724.66 | 610.54 | 92.06 | 3802.13 | 21.96 | 5.29 |
> > > | S | 5 | 994 | 69.88 | 0.37 | 1487.4 | 600.87 | 62.67 | 3727.26 | 26.95 | 4.01 |
> > > | M | 1 | 958.38 | 25 | 6.17 | 2256.49 | 283.37 | 100 | 4039.2 | 3.18 | 2.34 |
> > > | M | 3 | 1059.39 | 27.63 | 1.11 | 1739.41 | 619.66 | 92 | 3974.24 | 8.12 | 2.11 |
> > > | M | 5 | 1063.36 | 134.72 | 0.57 | 1505.21 | 601.2 | 56.07 | 3920.17 | 7.52 | 1.6 |
> > >
> > > * All experiments used Partial (which means 3 target modules).
> > > * In Setup, S/M denotes single/multiple LoRA; S/M in unified tasks denotes the number of LoRA in inference tasks, while fine-tuning tasks are all single LoRA.
> > > * tps denotes tokens per second, FTime/ETime denotes Fine-tuning/Evaluation Time, FTP/DTP denotes Fine-tuning/Decoding throughput, SLO denotes Service Level Objective (Attainment).
> > > * Configurations that appear in the paper but are not listed in the table above are not supported by FlexLLM. The issues with target modules have been identified in the paper, and the relevant experiments in the unified task can only perform single LoRA fine-tuning because FlexLLM cannot perform multi-LoRA fine-tuning.
> > > * FlexLLM will only execute one fine-tuning task at a time, even if two fine-tuning requests are submitted simultaneously; the fine-tuning instances of FlexLLM consume far more than 40G of GPU memory (H800 has 80G). This means that in our scenarios, similar to PEFT, FlexLLM can only run multiple fine-tuning requests sequentially. The data in the table is the result of sequential runs.
> > >
> > > The empirical results above are consistent with our previous findings, namely that Loquetier outperforms FlexLLM in these scenarios.
> > >
> > > We prioritized FlexLLM experiments in response to reviewers' concerns, but S-LoRA experiments are also underway. Based on previous proxy evaluations and the results above, we are even more confident that Loquetier outperforms the S-LoRA+PEFT baseline. We will complete the experiments as soon as possible and share the results.

---

> > > > ### Comment · Reviewer_fRyY · 2025-08-09
> > > > **Re: Rebuttal**
> > > >
> > > > Thank you very much for your additional comments and experimental results. I also greatly appreciate the effort taken to perform comparisons to FlexLLM - the ones currently presented in the response are much more convincing, and clearly support the claims about the efficiency of Loquetier. This, as well as the fact that Loquetier does tackle a setting that is not addressed adequately by previous frameworks, makes me willing to raise my score.

---

> > > > > ### Author Response · Authors · 2025-08-09
> > > > >
> > > > > Dear Reviewer fRyY and all other reviewers,
> > > > >
> > > > > Thank you very much for your thorough reviews and insightful comments. We, the authors of Loquetier, sincerely appreciate the time and effort you have dedicated to evaluating our work. Your feedback has been invaluable in helping us strengthen Loquetier, and we will update all the additional experiments in the next revision of the paper.

---

> > > ### Author Response · Authors · 2025-08-09
> > >
> > > Dear Reviewer fRyY and all other reviewers,
> > >
> > > Apologies for the disturbing. We would like to share the current status of our additional experiments related to S-LoRA. To reduce excessive email notifications, we are posting our response here only.
> > >
> > > Below are our newly obtained evaluation results related to S-LoRA in inference tasks:
> > >
> > > | | Single LoRA | | Multiple LoRA | |
> > > | - | - | - | - | - |
> > > | RPS | Decoding Throughput (tps) | SLO (%) | Decoding Throughput (tps) | SLO (%) |
> > > | 1 | 20.58 | 21.75 | 13.02 | 14.38 |
> > > | 2 | 22.31 | 23.25 | 14.21 | 14 |
> > > | 3 | 22.5 | 22.29 | 14.67 | 11.38 |
> > > | 4 | 22.47 | 22.28 | 15.01 | 9.44 |
> > > | 5 | 22.86 | 22.13 | 11.79 | 7.42 |
> > >
> > > * tps denotes token per second, SLO denotes Service Level Objective (Attainment).
> > >
> > > We would also like to provide a brief discussion:
> > >
> > > First of all, S-LoRA does not support the LLaMA 3 series models, and its repository has been archived. This limitation arises from the Group Query Attention (GQA) architecture used in LLaMA 3, where the shapes of K and V differ from those of Q and O. Consequently, the shape of the weight matrix B in the LoRA linear layers for K and V also differs from Q and O. Current S-LoRA requires all LoRA weights within the same layer to be concatenated at runtime. However, due to the shape discrepancies mentioned above, this concatenation operation fails. As a workaround, we replicate K and V weights in advance during model initialization to enable S-LoRA to start properly.
> > >
> > > Besides, S-LoRA supports applying LoRA only on the four linear layers Q, K, V, and O, and does not support the up, gate, and down layers within the MLP. Therefore, its runtime efficiency resembles the ***Partial*** scenario described in our paper, where only three linear layers (up, gate, down) are targeted.
> > >
> > > In our experiments, we observed instability in the S-LoRA kernel, which frequently produced incorrect outputs leading to NaN or Inf values. These errors propagate quickly, causing model generation failures. At this time, we did not modify the kernel. Our preliminary analysis suggests this may be due to missing synchronization mechanisms in some computational steps. (Note that this is an initial observation and may not be definitive.)
> > >
> > > Due to these issues, S-LoRA struggled to complete all inference requests in our scenarios, as it frequently outputs the eos token directly, making SLO appear better than it actually should be. For the requests that did succeed, decoding throughput (in single LoRA scenario) was approximately 22.14 tokens per second. Additionally, S-LoRA exhibited very low efficiency in merging pre-filling and decoding requests and removing completed requests, resulting in minimal benefits from batch processing.
> > >
> > > Lastly, S-LoRA requires over 35 GB of GPU memory during inference, which prevents concurrent fine-tuning and inference with PEFT on the same GPU.

---

### Note · Authors · 2025-08-12

Dear PCs, SACs, ACs, and Reviewers,

We sincerely thank the reviewers for their recognition of the contributions in this work and for their thoughtful, constructive suggestions. In the initial reviews, our submission was commended for presenting a clear and easy-to-understand paper and a solid, comprehensive framework (Reviewer fRyY); an architecture with promising prospects and scalability for LLMs (Reviewer WdRD); effective solutions to important problems, reproducible code, and a clearly structured, well-written manuscript (Reviewer tvxa); and an effective kernel design approach with a clean isolation mechanism, enhanced flexibility, and strong compatibility, thereby increasing the solution’s practical applicability and versatility (Reviewer VwoD).

During the rebuttal phase, we addressed the identified concerns and weaknesses with further clarifications and additional experiments:

- Clarifications
  - We expanded and emphasized the background, motivation, and objectives of our work, as well as the experimental settings, baselines, and metrics.
  - We provided a clearer design comparison for the Virtualized Module mechanism to better highlight our contributions.
- Additional evaluations:
  - Prior to writing the paper, we had already conducted substantial work to address limitations in FlexLLM for fine-tuning tasks, abnormal inference performance, and compatibility issues with S-LoRA for Llama 3 series models, as described in the original submission. In response to reviewer feedback, we performed more direct comparisons to highlight the advantages of our method more clearly, going beyond proxy evaluations and indirect inferences.
  - Despite the presence of issues in the default repositories of FlexLLM and S-LoRA, we evaluated modified versions of both baselines, while ensuring their original performance was unaffected. Based on the results, we could found that Loquetier outperforms them in inference, fine-tuning, and unified tasks.
  - We further evaluated Loquetier under simulated dynamic workloads using the BurstGPT dataset, enhancing realism and representativeness, and demonstrating clear advantages in real-world scenarios.
All supplementary experiments and results have been incorporated into our internal manuscript and will appear in the next version of the paper.

We once again extend our sincere gratitude to the reviewers for their constructive feedback, which has helped us further strengthen and refine our work.

---

### Decision · Program_Chairs · 2025-09-17

**Decision:**

Accept (poster)

**Comment:**

The paper presents Loquetier, a virtualized multi-LoRA framework that unifies fine-tuning and serving. It features virtualized modules that isolate model modifications, along with an optimized computation flow and kernel design that merges training and inference paths. The authors conduct an empirical study using three workloads and demonstrate that Loquetier consistently outperforms existing baselines.

During the rebuttal, the authors provide a comprehensive response, including new experiments, to address the reviewers' concerns. The reviewers also acknowledged the response. The AC believes this paper makes a valuable contribution to the field, so recommends acceptance.